# Up-to-fivefold reverberating waves through the Earth's center and distinctly anisotropic innermost inner core

Thanh-Son Phạm [1] ✉ & Hrvoje Tkalčić [1]

Probing the Earth's center is critical for understanding planetary formation and evolution. However, geophysical inferences have been challenging due to the lack of seismological probes sensitive to the Earth's center. Here, by stacking waveforms recorded by a growing number of global seismic stations, we observe up-to-fivefold reverberating waves from selected earthquakes along the Earth's diameter. Differential travel times of these exotic arrival pairs, hitherto unreported in seismological literature, complement and improve currently available information. The inferred transversely isotropic inner-core model contains a ~650-km thick innermost ball with P-wave speeds ~4% slower at ~50° from the Earth's rotation axis. In contrast, the inner core's outer shell displays much weaker anisotropy with the slowest direction in the equatorial plane. Our findings strengthen the evidence for an anisotropically-distinctive innermost inner core and its transition to a weakly anisotropic outer shell, which could be a fossilized record of a significant global event from the past.

Earth's inner core (IC), which accounts for less than 1% of the Earth's volume, is a time capsule of our planet's history[1,2]. As the IC grows, the latent heat and light elements released by the solidification process drive the convection of the liquid outer core[3,4], which, in turn, maintains the geodynamo. Although the geomagnetic field might have preceded the IC's birth[5], detectable changes in the IC's structures with depth could signify shifts in the geomagnetic field's operation, which could have profoundly influenced the Earth's evolution and its ecosystem[1,6]. Therefore, probing the innermost part of the IC is critical to further disentangling the time capsule and understanding Earth's evolution in the distant past.

In the first several decades after its discovery[7], seismological investigations of the IC focused mainly on the characterization of its isotropic structure and boundary with the liquid outer core[8,9]. However, since the 1980s, the studies of its anisotropic structures have complemented the existing knowledge. P-wave transverse isotropy, specifically the IC bulk's depth-independent cylindrical anisotropy, was the first proposed conjecture to explain the travel times of compressional body waves traversing the IC (PKIKP waves)[10,11] and Earth's normal-mode splitting functions[12]. However, that conjecture was soon updated by the discoveries of anisotropy's hemispherical dichotomy[13]

and radial variations[1,14,15]. Recent studies tend to introduce more complex structures, including variations in P-wave anisotropy[16,17] and attenuation[18]. Additionally, S-wave anisotropy has also been recently observed[19,20]. The seismological observations have provided essential constraints on the mineralogical properties of stable crystallographic structures of iron in the IC. However, it remains an open debate whether the hexagonal close-packed (hcp)[21,22] or body-centered cubic (bcc)[23] phase of iron stabilizes at the IC temperature-pressure conditions.

Despite the expanding number of studies, the IC remains enigmatic, particularly in its innermost part. That is because of the inherent limitation in a volumetric sampling of the existing seismological probes and the fact that this Earth's volume is buried beneath other layers. On the one hand, travel times and amplitudes of PKIKP waves have been the primary short-period tools to obtain inferences on spatially-distributed properties such as anisotropy and attenuation[24–26]. However, to probe the centermost ball of the IC, seismic stations and earthquakes must be positioned at near antipodal distances, which is challenging in practice due to the confinement of large subduction-zone earthquakes in the quasi-equatorial belt and the limited seismic deployments in the oceans and remote areas. On the

[1]Research School of Earth Sciences, The Australian National University, Canberra, ACT, Australia. ✉e-mail: thanhson.pham@anu.edu.au

other hand, normal modes have limited lateral and radial resolution because of their long-period nature, and their sensitivity approaches zero in the Earth's center.

To improve the spatial sampling of the Earth's deep interior, coda correlation studies[27–29], which exploit correlated features lasting in long earthquake recordings, have emerged as promising tools to probe the Earth's interiors. The correlation wavefield that exploits the similarity of weak signals samples the IC differently from the previous techniques[19,30] (for a recent review, see ref. [31]). Most recently, the correlation feature I2* has been suggested as another class of observations to probe the P-wave anisotropy of the IC[32]. The challenges in proceeding with this correlation approach include the overwhelmingly complex correlation features kernels and require future investigation.

The innermost inner core (IMIC) was initially hypothesized as a central ball within the Earth's IC characterized by distinctive anisotropic properties from the outer shell[1,33]. The original studies[1,33] suggested the 300-km-radius IMIC with a slow direction at ~45° from the fast axis, aligning with the Earth's rotation axis. This hypothesis has been corroborated by subsequent studies using the International Seismological Center (ISC) datasets via robust non-linear searches[34], dedicated picking of the antipodal PKIKP waves[35–37], and normal mode analysis[15]. However, there are still significant unknowns related to the IMIC radius, the nature of the transition to the outer IC, and its precise anisotropic properties, such as the strength and the fast and slow directions. These topics keep inspiring further investigations.

This study reports a previously unobserved and unutilized class of seismological observations of reverberating waves through the bulk of

the Earth along its diameter up to five times, later referred to as PKIKP multiples. To our knowledge, reverberations from more than two passages are hitherto unreported in the seismological literature. Simultaneous observations of these exotic arrivals at regionally dense seismic networks opportunistically provide tools to constrain the IMIC properties because they sample the IMIC in an unprecedented fashion. Measurements from these observations, in agreement with recent independent findings[32,34,37,38], confirm an anisotropically distinctive IMIC from the less anisotropic outer shell.

## Results

### Multiple body-wave reverberations through the Earth's inner core

Stacking seismic waveforms from multiple seismic stations can enhance weak but coherent seismic signals while suppressing incoherent noise. The stacked waveforms can be gathered in a two-dimensional image of lapse time and epicentral distance to represent the seismic wavefield with spatial coherency. In the 1990s, due to the sparsity of global seismic stations, waveforms from many earthquakes were gathered to produce a global stack covering a complete range of epicentral distances[39,40], 0–180°. The stacks have served as a reliable tool for identifying many weak seismic arrivals, for example, those relating to the mantle discontinuities[40] and in the sound fixing and ranging (SOFAR) channel[41] due to the variation of temperature and salinity of seawater with depths in the ocean.

This study uses the ever-growing global seismograph network (Fig. 1A) to produce global stacks for some significant seismic events

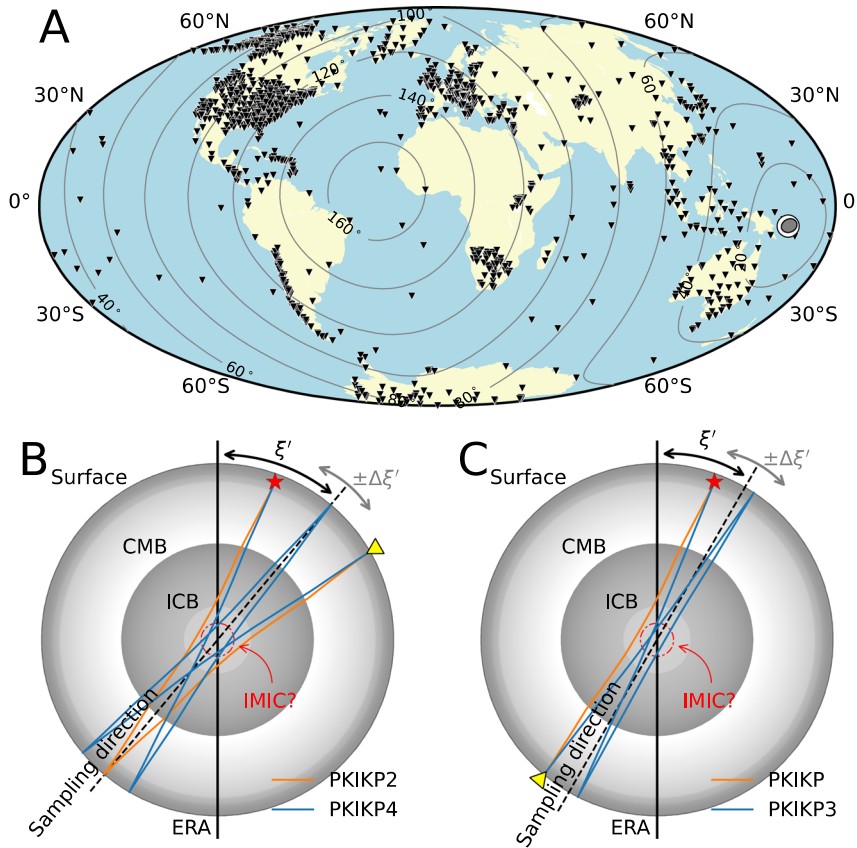

**Fig. 1 | Location map and schematic ray paths of PKIKP multiples. A** The location map of the 22 Jan 2017, Mw 7.9 Solomon Islands earthquake and stations that contribute to the global stack (see Fig. 2). Black inverted triangles denote the seismic stations, and beachball marks the location and mechanism of the earthquake. The contours show the epicentral distances from the event. **B** Schematic ray paths of the second and fourth PKIKP multiples, i.e., PKIKP2 and PKIKP4, reverberating along the Earth's diameter twice and four times. $\xi' \pm \Delta\xi'$ is the representative sampling direction angle relative to the Earth's rotation axis (ERA). The red dashed-dotted circle denotes the tentative innermost inner core (IMIC) boundary with a radius of 650 km. **C** Similar to panel (B) but for PKIKP and the third multiples, i.e., PKIKP3 reverberating along the Earth's diameter three times.

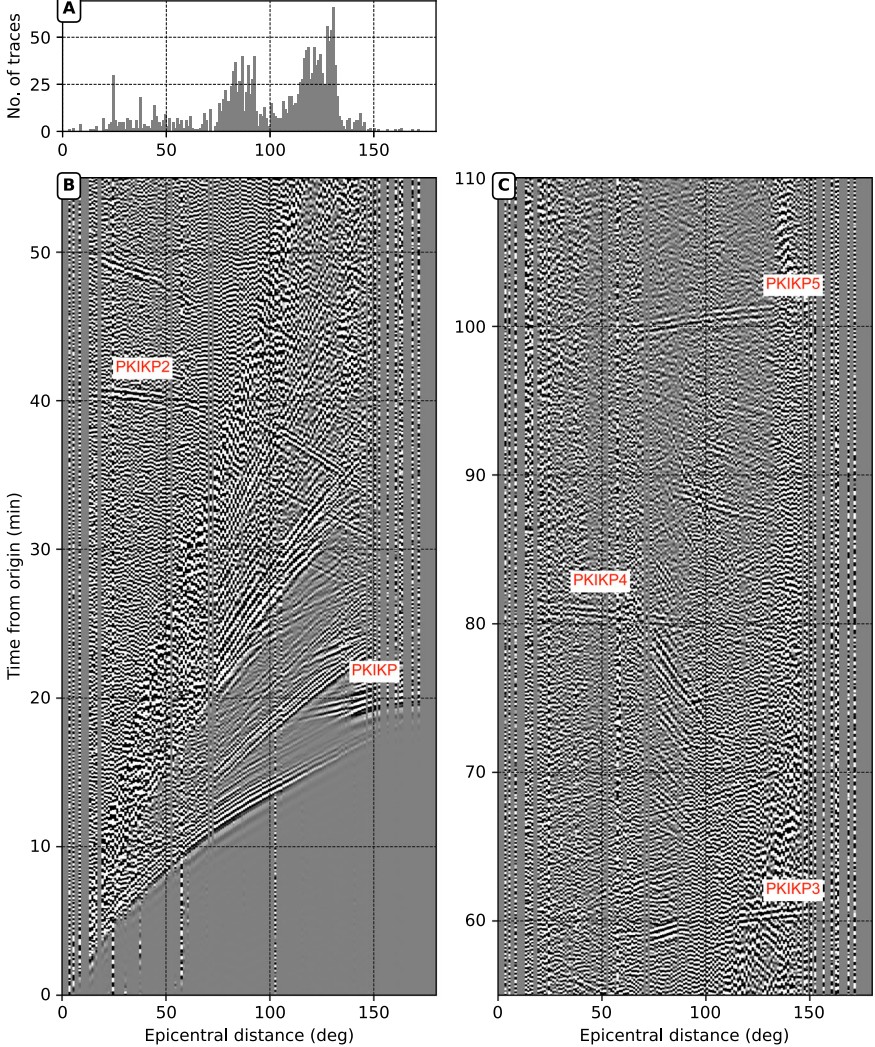

**Fig. 2 | Global stack for the 22 Jan 2017, Mw = 7.9 Solomon Islands earthquake.** **A** Histogram of seismic waveforms as a function of epicentral distance in 1-degree bins. **B** and **C** Global stacks spanning 0–55 min and 55–110 min. Exotic podal and antipodal reverberations up to five multiples (with near-horizontal slope) are labeled by red fonts.

individually (Fig. 2). We retrieve seismic waveforms from several international data centers (i.e., IRIS, ORFEUS, GFZ, ETHZ, and INGV) to construct global seismic stacks with 1-degree distance bins for all large earthquakes (Mw 6.0+) in the last decade (see Methods section). To avoid possible misalignment of phases due to heterogeneities of the Earth, all waveforms are bandpass filtered at long periods between 10–100 s. For example, Fig. 2 shows a global stack of a thrust-faulting earthquake (22 Jan 2017, Mw 7.9) in the Solomon Islands (see Fig. 1A for a location map). In the stack, the horizontal axis spans epicentral distances from 0°, meaning earthquake and station are nearly collocated, to 180°, meaning they are antipodal.

Perhaps the most eye-catching features observed in the global stack of the 2017 Solomon Islands event (Fig. 2) are relatively flat arrivals indicating that the associated seismic phases arrive steeply to the Earth's surface. Their arrival times and slowness properties suggest they are seismic phases reverberating along the entire Earth's diameter, including the inner core, multiple times. Thus, we adopt an abbreviated nomenclature, similar to previous counterparts in the correlation wavefield[29]: PKIKP (I), PKIKP2 (I2), PKIKP3 (I3), PKIKP4 (I4), and PKIKP5 (I5), in which the last digits represent the number of passages reverberating the entire Earth's diameter as compressional waves (see their schematic ray paths Fig. 1B, C). Observations of such exotic arrivals in several other single-event stacks are included in the

supplementary material (Fig. S1). They present similar quality exotic phases of three- or fourfold reverberations that can be routinely observed. Fivefold reverberations (as in Fig. 2) are seen clearly on only a few global stacks.

Apart from routine antipodal single-passage PKIKP observations, the podal PKIKP2 waves have been observed on individual and array elements[42]. However, to our knowledge, PKIKP3, PKIKP4, and PKIKP5 waves have not been reported. It is worth mentioning that the observation of multiple podal and antipodal reverberations had been absent in the past global stacks, possibly because data from many events must have been employed[43] to construct a stack of similar quality to Fig. 2 due to the limited numbers of seismic stations existing in the past. The events were often confined to depths shallower than 50 km to avoid the mismatch due to the timing variation of earthquake depths[39]. However, the mixture of many events adversely impacted the expression of the multiple reverberations because they exist in only a tiny fraction of all events. Therefore, stacking all events resulted in destructive interference leading to the absence of the exotic reverberations.

The unambiguous expressions of the multiple podal and antipodal reverberations can be attributed to two factors. Firstly, events inducing the multiple reverberations have relatively large magnitudes (Mw >6.0) but a simple energy release in either thrust or normal

faulting mechanisms. The properties were favorable to emitting significant seismic energy vertically downward and efficiently illuminating the Earth's interior[44]. Secondly, the corridors traversing through the Earth's inner core are nearly transparent for P-wave propagation at the steep incidence because reflection coefficients[45] at major internal interfaces, such as the core-mantle and the inner-core boundaries, are small for near-vertical incidence (Fig. S2). Additionally, the attenuation effect for ~10 s periods is weak in the IC's upper part (Fig. S3), which is believed to be the most attenuative region in the Earth's interior[24]. The combination of these favorable conditions helps sustain significant energy for the PKIKP multiple passages through the Earth's bulk.

There are features in the global coda correlograms that look similar to PKIKP multiples and were explained to arise due to the similarity of late, weak arrivals after large earthquakes[29]. The emergence of the core-sensitive signals in the coda-correlation wavefield thus inspired us to search for exotic reverberations in the direct seismic wavefield that results in the correlation features' formation. Unlike the exotic correlation features, whose complex geometrical sensitivity kernels to Earth's internal structures must yet be fully understood[46], the observed multiple PKIKP waves are practical because their sensitivity can be mapped along their ray paths. Here, we applied these observations to constrain the IMIC.

## Constraint of Distinct Anisotropy in the Innermost Inner Core

As the podal (receivers near the source) and antipodal (receivers on the antipodal side from the source) waves spend multiple passages through Earth's inner core, their travel times can provide constraints on the IC structures once they are corrected for contributions of source location errors, the Earth's ellipticity, and mantle heterogeneities. In the Methods section, we describe a procedure to measure the travel time residuals for pairs of exotic arrivals, such as PKIKP4-PKIKP2 and PKIKP3-PKIKP, on the stacked waveforms over dense regional seismic networks. In our regional observations, we can use a slightly shorter period band (i.e., 7–13 s; see Methods section) than in the global stacks (i.e., 10–100 s in Figs. 2 and S1), as the propagation paths to array elements are likely to experience fewer heterogeneities in the mantle. Occasionally, even shorter period observations could be made (e.g., 1–10 s for the 2018 Anchorage earthquake). However, the 7–13 s period band was deemed suitable for obtaining as many as 16 differential travel time measurements of the PKIKP multiple pairs (see Methods section). Thus, we proceed with this period band in the subsequent analysis. Supplementary Methods present finite-frequency numerical experiments using the spectral element method[47] in a 2D Earth section to demonstrate the feasibility and measurement sensitivity of the 7–13 s PKIKP multiples in probing a 650 km-radius IMIC.

Fig. 3 considers a cylindrically anisotropic model of the Earth's inner core to fit the residual travel time residuals relative to the ak135 reference model[48], corrected for mantle heterogeneities using the DETOX-P3[49] mantle model for compressional waves. In the cylindrical model, the relatively small perturbation from the background velocity of the Earth's IC is expressed as a function of a single dependent parameter, the sampling angle, $\xi'$; specifically[50],

$$\frac{\Delta v}{v} = \gamma + \varepsilon \cos^2\xi' + \sigma \sin^2\xi' \cos^2\xi'. \tag{1}$$

In Eq. 1, $\varepsilon$ and $\sigma$ are the controlling parameters of the model, and the baseline shift $\gamma$ accounts for the uncertainty of the 1D reference model. Equation 1 is thus a quadratic function of $\cos^2\xi'$. In this equation, a representative sampling angle, $\xi'$, is defined to represent the sampling direction of the exotic arrival pairs relative to the Earth's rotation axis (see more details in the Methods section). We estimate the parameters for two anisotropic models of the Earth's IC, including (i) the bulk IC model assuming the directional dependence of seismic wave speed does not change with depths and (ii) the model with an

IMIC, consisting of two anisotropically different domains, a concentric outer shell, and an innermost ball, i.e., IMIC. In the model with an IMIC, the anisotropic strength and depth extent, $H$, of the outer layer from the inner core boundary are fixed to the recent model proposed by ref. [34], in which $H = 650$ km, $\varepsilon = 1.45\%$, $\sigma = -1.07\%$, and $\gamma = 0$, as our data cannot independently constrain the outer layer's parameters.

In this study, we use the orthogonal distance regression method[51] to estimate the anisotropic parameters (see the Methods section) because this method can account for the measurement uncertainties of both explanatory variable (i.e., $\cos^2\xi'$) and response variable (i.e., $\Delta v/v$) of Eq. 1, which are both significant in our observed data. The mean anisotropy models are plotted in Fig. 3 in dark solid lines. Light blue lines represent the models' uncertainty by simulating the parameters with correlated uncertainty using the Monte Carlo method. The bulk IC models indicate that P-wave travels through the IC at the lowest speed at around $\xi' \approx 48°$, while the speed is ~2.8% faster along the polar direction and 1.7% faster along the equatorial plane (Fig. 3A). In the two-layer IC model, the slow direction in the IMIC remains almost unchanged, but P-waves are ~4.0% and 3.4% faster when traveling along the polar direction and equatorial plane (Fig. 3B). Similar models with the slow direction offset significantly from the ERA are also observed when other mantle models are used to correct for mantle heterogeneities, i.e., MIT-P08[52] in Fig. S6, LLNL-G3Dv3[53] in Fig. S7, and no mantle correction applied in Fig. S8. Thus, we consider the changing patterns of P-wave anisotropy with depths in the IC to be robustly observed.

Note that our measurements sample near the Earth's center and have minimal depth sensitivity, so we cannot directly favor the bulk IC model (Fig. 3A, B) or the IC model with an IMIC (Fig. 3C, D) based on the data fits alone. Instead, the centermost-sensitive observations demonstrate a prominent anisotropic pattern in the IMIC (Fig. 3B) with slow directions residing at mid-range latitudes. This property, in line with pioneering observations[1,33] drives the anisotropic pattern in the bulk IC models (Fig. 3A) because the anisotropy in the outer IC (OIC) is weaker than the IMIC with slow directions around the equatorial plane[2] (a representative anisotropic model of OIC from ref. [34] is plotted in Fig. 3B). This enables us to infer a distinctively anisotropic IMIC from the datasets.

## Discussion

The direct observations of PKIKP multiples using regional seismic arrays equip seismologists with additional seismic phases to sample the center of the Earth's IC. This approach has clear advantages even with the existing distribution of earthquakes and seismograph networks. Firstly, the observed reverberations provide a unique sampling style of the Earth's IC along the north-south direction. Although this direction has been sampled with the South Sandwich Islands (SSI) events recorded in Alaska[54–57], due to the epicentral distance range, only the outer parts of the IC were sampled. However, our observation of the exotic phases from the 2018 Anchorage event, recorded by the elements by the Alaskan branch of the US Transportable Array (Fig. 5) within the 10° epicentral distance range now sample the very center of the IC due to their unique podal geometry (see ray paths in Fig. S9).

Secondly, the uncertainties due to earthquake location errors are mitigated by measuring the differential travel times of a pair of exotic phases. The impacts both Alaskan and SSI slabs might have on the travel times are mitigated for the measurements associated with the three SSI events in 2018 (see their location in Fig. S4) thanks to the proximity of PKIKP and PKIKP3 ray paths at both source and receiver sides (see Fig. 1C). When the stations and events are restricted to the podal and antipodal configurations (i.e., <50° for PKIKP4-PKIKP2 and >155° for PKIKP3-PKIKP), the PKIKP reverberating arrivals sample the centermost 650 km of the IC several times (see Fig. 1B, C), which amplifies the evidence for any travel-time anomalies.

One of the challenges in expanding the use of the exotic PKIKP multiples is the involvement of large, dense seismic arrays such as

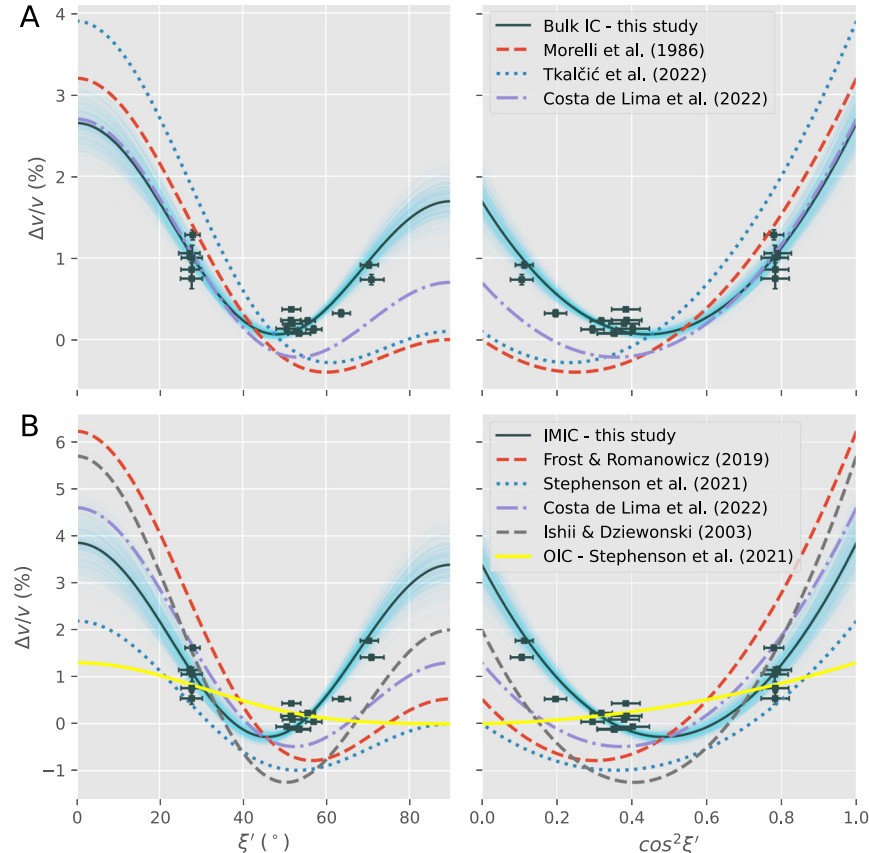

**Fig. 3 | Cylindrically anisotropic model of Earth's inner core (IC) inferred from exotic PKIKP multiples' travel times.** Fractional velocity (see the Methods section) and fitting curves are plotted as a function of $\xi'$, the representative sampling direction of the ray (left) (see Fig. 1), and $\cos^2\xi'$ (right). All measured differential travel times are corrected for mantle heterogeneities using the DETOX_P3 model[49]. Associated uncertainties are plotted by error bars. Dark solid lines are the optimal anisotropic models parameterized in Eq. 1, and light blue opaque lines represent the uncertainty surrounding the optimal model. Various broken lines show models from previous studies (see the legend). The top row (**A**) compares our inferred bulk IC model with other models, while the bottom row (**B**) compares our inferred innermost inner core (IMIC) model with other models. Yellow lines in (B) show a representative anisotropy model of the outer inner core (OIC) taken from ref. [34], which is used to account for the OIC structure and obtain our IMIC models.

USArray or AlpArray. The Earth's heterogeneous structures beneath large arrays require observations at longer periods, leading to broad sensitivity to the IC. Due to limited data access, this study has not explored the large-scale ChinArray. Overall, it will be challenging to introduce brand new sampling directions to the IC unless other large-array projects get underway in the next several decades. Thus, future attempts in this direction should focus on identifying higher-frequency observations using smaller aperture networks, e.g., down to national-scale and/or local seismograph networks, which are more available worldwide.

As shown in Fig. 3, this study's main findings of IC's cylindrically anisotropic models with the slowest direction at $\xi' \approx 48°$ (Fig. 3B) are consistent with previous studies concerning the IMIC, such as the comprehensive absolute PKIKP datasets released by the ISC[1,34], dedicated handpicked datasets[35–37] (Fig. 4B), and constraints from the global correlation wavefield[32] (Fig. 4C). Both the *bcc* crystallographic structure of iron[23] and *hcp* iron[36,58] can have slow directions at oblique angles relative to the ERA, depending on the orientation of iron crystals, which agrees with our results. Although the *hcp* iron enables more approachable studies, recent ab initio calculations at the IC temperature and pressure conditions suggest that *bcc* crystals[59,60] could inherently explain the reduced shear modulus of the Earth's IC, high anisotropy, high Poisson's ratio, and high attenuation.

The properties in the innermost 650-km shell of the IC are significantly different from the outer shell (Fig. 3A), characterized by weak anisotropy with the fast axis along the Earth's rotation axis and a slow direction residing in the equatorial plane[54,61] (see the schematic representation of this anisotropic pattern in Fig. 4A). Note that a hemispherical structure of the IMIC has been suggested in two recent studies[38,62]. When inverting for the 3D structures of the IC using single-passage PKIKP probes, the former study[38] found an IMIC confined in the eastern hemisphere with a slow direction at -55 ± 16°. Our data sampling near the planetary center yields a similar value for the slow propagation direction. Several geodynamical models have been invoked to explain the changes in the anisotropic properties with depth, including (1) diminishing strength of thermal convection over time[63]; (2) preferential crystallization due to the transition in the deformation pattern over time coupled with density stratification[64]; and (3) the IC's growth could have been conditioned by the sedimentation of light elements at the ICB, which is linked to chemical variations in the outer core[65,66].

In conclusion, we have employed the modern global network to compute stacks of seismic wavefields induced by individual earthquakes. We report robust observations of podal and antipodal reverberations of compressional waves through the Earth's bulk. Opportunistically, dense networks at continental scales such as the USArray, including mainland and Alaska deployments, or the AlpArray in Europe are exploited to sample the Earth's center by measuring the differential residual between pairs of exotic arrivals. The inferred model supports the existence of the anisotropically-distinctive IMIC from its outer shell, which might indicate a fundamental shift in the IC's growth regime in the Earth's past. We now have enough seismological

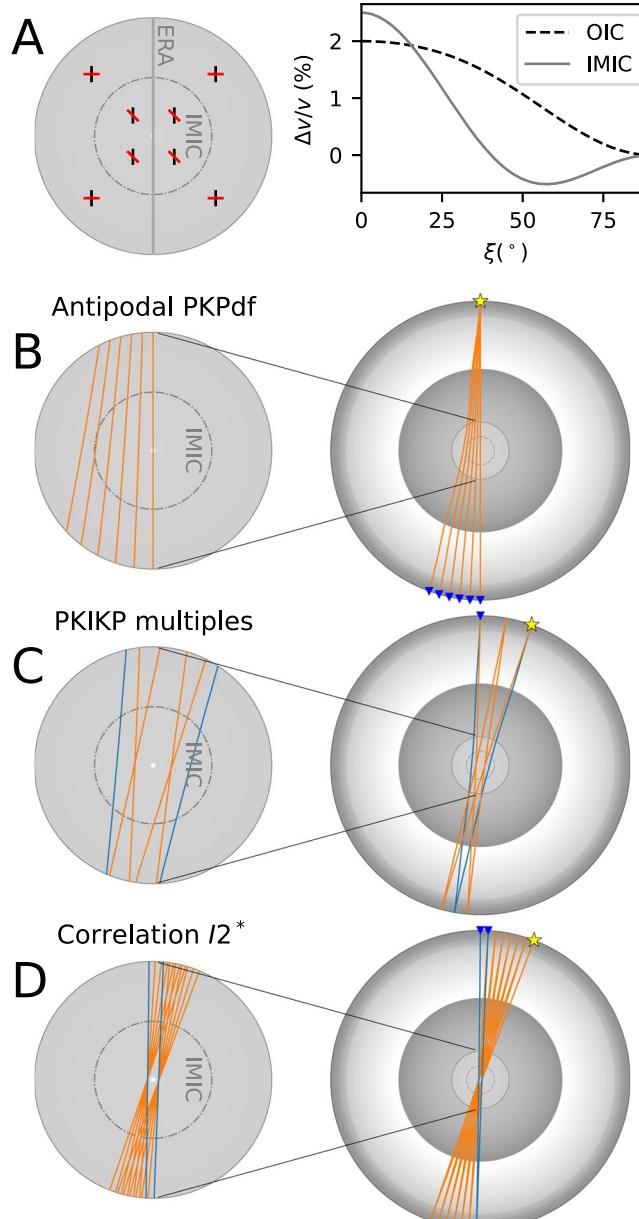

**Fig. 4 | Schematic model of the inner core (IC) containing an innermost inner core (IMIC) and various body-wave probing methods to the Earth's center.**
**A** On the left panel, the IC model contains IMIC with two distinct P-wave transversely isotropic patterns in outer inner core (OIC) and IMIC; black and red bars represent the fast and slow anisotropic direction; a vertical gray line represents the Earth's rotation axis (ERA). On the right panel are cylindrically anisotropic models of OIC (black dashed line) and IMIC (gray line). Schematic views of IMIC-sampling methods using (**B**) absolute PKIKP waves, (**C**) PKIKP multiples (this study) – blue and orange ray lines represent ray paths of the second and fourth PKIKP multiples, and (**D**) correlation feature I2* at the two receivers (blue ray paths) results from cross-correlating high-multiples of PKIKP (orange ray paths). The cross-sections of the Earth in the right column contain ray path segments sampling the OIC and IMIC. Yellow stars are sources, and blue inverted triangles are receivers.

evidence from several different lines of investigation about the existence of IMIC. Future efforts should be directed toward characterizing the IMIC-OIC transition (its depth and nature). The findings reported here are a consequence of the unprecedentedly growing volume of digital waveform data and will hopefully inspire further scrutiny of existing seismic records for revealing hidden signals that shed light on the Earth's deep interior.

## Methods

### Data retrieval and pre-processing

Seismic waveforms are gathered from multiple data centers, including IRIS, ORFEUS, ETH, INGV, and GFZ. Data from four later centers improve coverage in Europe, mainly used in the detection pair of PKIKP3-PKIKP for events in the Kermadec Islands region. We only use data from stations that continuously record for 120 min from the event origin time. All retrieved seismic waveforms are corrected for instrumental responses to obtain velocity seismograms and resampled to 10 samples per second (sps) in the preparation stage. Thus $N = 72,000$ (7200 s × 10 sps) is the number of points in each waveform. All data management and processing tasks are performed using the obspyDMT[67] and ObsPy[68] packages.

### Construction of global stack

Here, we describe the procedure to construct the global stack of the direct wavefield. After correcting for the instrumental response, seismic records are bandpass filtered between 10–100 s (zero phases, three corners), then grouped in 1-degree distance bins. Next, we deployed a median filter to remove seismic traces with anomalous amplitudes due to possible instrument malfunctions or glitches. Particularly, the median value of maximum absolute amplitudes for a distance bin is obtained,

$$v_{med} = med_{k=1,M} \left( \max_{i=1,N} |v_i^k| \right) \quad (2)$$

where $M$ is the number of traces in a bin and $N = 72,000$ is the number of samples in time. Any waveform trace having its maximum absolute amplitude, $\max_{i=1,N} |v_i^k|$, larger than five times the median value, $v_{med}$, is discarded from the further processing. The factor of five was chosen empirically because anomalous records were efficiently rejected during a visual inspection for some events. There are no other measurements applied for quality control. The remaining traces in the bin are summed (i.e., linearly stacked) to render one vertical stripe in the 2D global stacks (e.g., Figs. 2 and S1). Because all waveforms are from a common earthquake, we do not apply any amplitude normalization, which alters the relative amplitudes of the waveforms. Instead, to improve the visibility of arrivals at significant lapse times like PKIKP4 and PKIKP5 in Fig. 2, we multiply all binned stacks by a common polynomial of elapsed time, $f(t) = (t \times 10)^4$. We use linear interpolation in visualizing global stacks in Figs. 2 and S1.

### Measuring residual travel times of exotic multiples

To measure the residual times for the exotic pair, we initially align individual waveforms according to theoretical predictions of PKIKP(n) and PKIKP(n + 2) (n = 1, 2) according to the *ak135* reference model[48] and corrected for the Earth's ellipticity[69] and mantle heterogeneities using several P-wave 3D mantle models[49,52,53]. It is worth noting that in the measuring procedure, stacking is necessary because the arrivals on individual waveforms have low signal-to-noise ratios, especially for the third and higher multiples. Fig. 5 shows data from ~350 vertical seismograms in the Alaskan branch of the US Array recording the 30 Nov 2018 Mw 7.1 Anchorage earthquake, where the DETOX-P3 model[49] is used to correct for mantle heterogeneities. The time corrections are applied to shift the waveforms accordingly (Fig. 5A, B), and they are then stacked to enhance signal-to-noise ratios (Fig. 5C, D). The stacked waveforms over the entire array show prominent signals of remarkable similarity for the two late arrivals, so the residual of the differential travel time of the arrival pair to the reference model can be determined using the cross-correlation:

$$\Delta t = (t_{PKIKP4} - t_{PKIKP2})_{obs} - (t_{PKIKP4} - t_{PKIKP2})_{pred}. \quad (3)$$

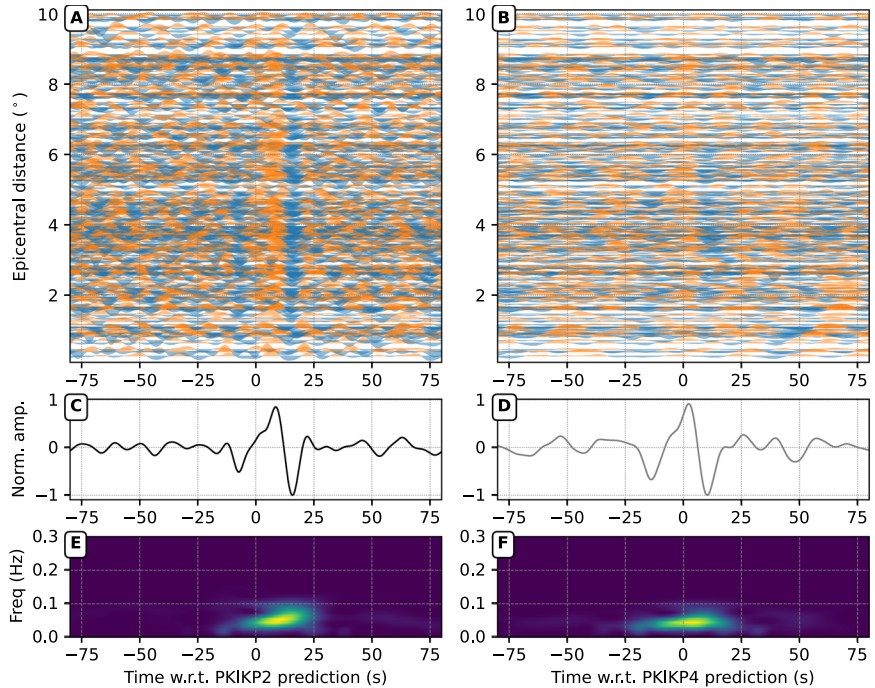

**Fig. 5 | Observations of second and fourth PKIKP multiples, i.e., PKIKP2 and PKIKP4, respectively, in the seismic wavefield from the 30 Nov 2018, Mw 7.1 Anchorage earthquake. A** Seismic records from the Alaskan network are aligned with the predictions of PKIKP2 arrivals, corrected by the Earth's ellipticity[69]. The waveforms are bandpass filtered in the period band of 7–13 s. **C** Linear stack of individual waveforms. **E** The spectrograms of stacked waveforms before filtering show the frequency content variation as a function of time. (**B**, **D**, **F**) Similar to (A), (C), and (E) but for the PKIKP4 arrivals.

We use the bootstrap method (Fig. S9) to estimate the uncertainty of the residual measurements for the network configuration with 5000 shuffles of the stations in the stacked waveforms with replacements, whose mean and standard variation are measured (shown on top of all panels of Fig. 6).

The measured travel time residuals are then converted into relative perturbations from the background velocity,

$$\frac{\Delta v}{v} = -\frac{\Delta t}{\left(\tau_{PKIKP(n+2)} - \tau_{PKIKP(n)}\right)_{pred}}. \tag{4}$$

In Eq. 4, $\tau_{PKIKP(n+2)} - \tau_{PKIKP(n)}$ are the theoretically predicted differences of propagating times, $\tau_{PKIKP(n)}$ $(1 \leq n \leq 4)$, of the IC segments in the PKIKP multiple ray paths, and $\Delta t$ is the measured residual. Thus, the velocity perturbation, $\Delta v/v$, defined in this fashion, is compatible with similar quantities inferred for PKP waves traditionally used in other inner core studies.

To retain the slight variation in the direction dependence of velocity perturbation, we only collect measurements of the PKIKP4-PKIKP2 residuals for the podal configuration, where epicentral distances are smaller than 50°, and the PKIKP3-PKIKP residuals for the antipodal configuration, where epicentral distances are larger than 155°. These distance criteria help reduce the variety of $\xi$-angles for individual sampling legs of higher multiple arrivals (see Fig. 1B, C). This is somewhat similar to the approach implemented by Costa de Lima et al.[32], where they used the travel times of the I2* correlation feature, manifested in the correlation of much higher-order reverberating PKIKP waves in late earthquake coda, to constrain the Earth's inner core anisotropy. Because of these criteria, we retain differential residual measurements for 16 events recorded by regional seismic networks in Alaska, the mainland United States, and Europe (see Fig. 6 for their stacked waveforms and residual measurements and Fig. S4 for their location maps).

We use a representative angle, $\xi'$, to represent the sampling direction of an exotic pair to the Earth's rotation axis (ERA). For the PKIKP4-PKIKP2 pair, $\xi' = \frac{\xi_1 + \xi_2}{2}$, where $\xi_1$ and $\xi_2$ are the angles to the ERA of the forward and backward PKIKP2's legs (see Fig. 1B). Similarly, for the PKIKP3-PKIKP, the $\xi'$ is the angle relative to the ERA of PKIKP (see Fig. 1C). The distance criteria mentioned above ensure that the most significant deviation of the representative angle $\xi'$ to any of the individual PKIKP legs in the multiples is less than 25° for PKIKP or PKIKP2 and less than 12.5° for PKIKP3 and PKIKP4. Furthermore, because the residuals are measured on stacked waveforms over a seismic array, the variation of $\xi'$ for all elements must be considered as the uncertainty of the sampling direction. We calculate the mean and standard deviation of $\cos^2\xi'$ instead of angle $\xi'$, because Eq. 1 is a quadratic function of $\cos^2\xi'$ and it is mathematically convenient when estimating its parameters,

$$\overline{\cos^2\xi'} = \frac{1}{N}\sum_{i=1}^{N}\cos^2\xi'_i;$$

$$\Delta\cos^2\xi' = \sqrt{\frac{1}{N-1}\sum_{i=1}^{N}\left(\cos^2\xi'_i - \overline{\cos^2\xi'}\right)^2}. \tag{5}$$

The subscript $i$ denotes individual array elements. The measured velocity perturbations ($\Delta v/v$), sampling direction ($\cos^2\xi'$), and their associated uncertainties are plotted as dark squares with error bars in Fig. 3. Note that the representative uncertainty of the sampling direction, $\Delta\xi'$, corresponds to the estimated uncertainty, $\Delta\cos^2\xi'$.

**Estimate of anisotropic model**

In this study, we use the orthogonal distance regression method[51] to estimate the anisotropic parameters (Eq. 1) because this method can account for the measurement uncertainties of both explanatory, $\cos^2\xi'$, and response, $\Delta v/v$, values, which are significant in our observed data. The outputs consist of the three anisotropic

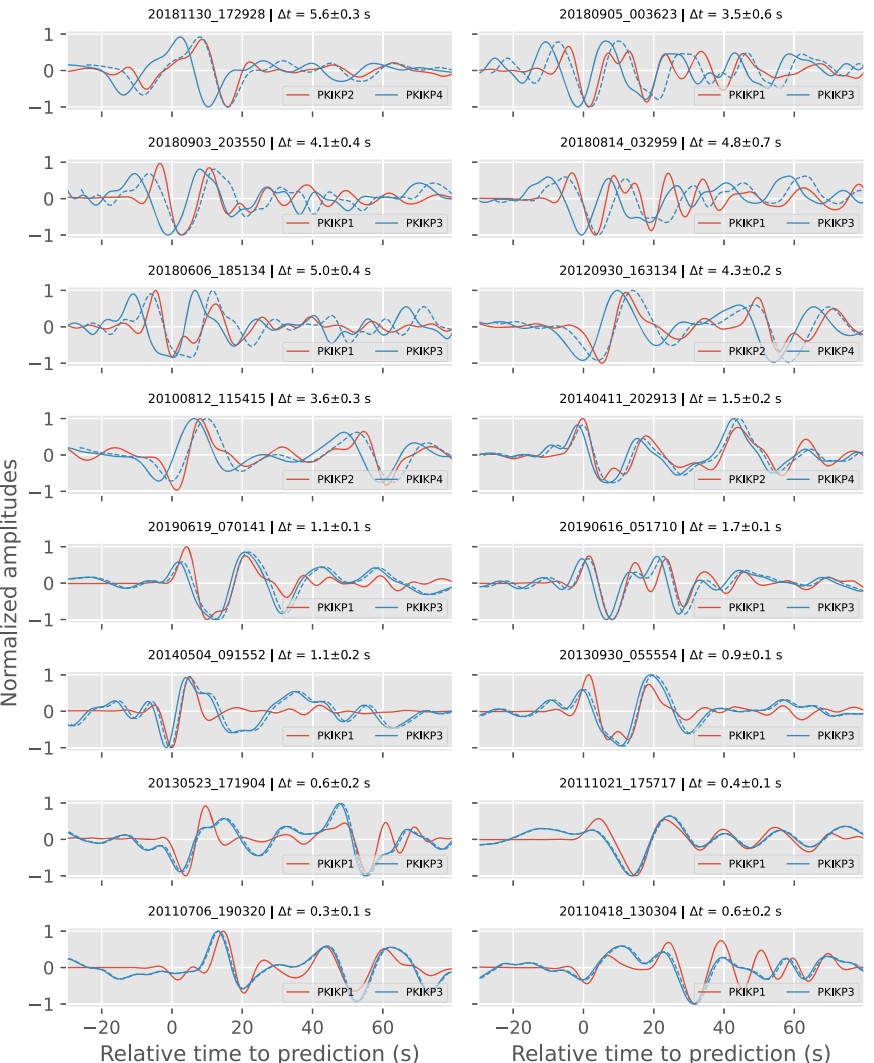

**Fig. 6 | Measurements of travel time residuals by cross-correlating stacked waveforms.** In each panel, solid lines are linearly stacked waveforms of the exotic pairs of PKIKP multiples. Dashed lines show cross-correlated and shifted original waveforms. Panel titles show the origin times and the measured residuals with bootstrapped uncertainties.

parameters' mean values and their correlated uncertainty in the form of a 3 × 3 symmetric covariance matrix. When mantle heterogeneities are corrected for using the DETOX-P3 model[49], estimated values for the bulk inner core model parameters are:

$$
\begin{array}{l}
\varepsilon \\
\sigma = \\
\gamma
\end{array}
\begin{array}{l}
1.0 \\
-8.3 \text{ and } C = \\
1.7
\end{array}
\begin{array}{ccc}
0.036 & -0.018 & -0.009 \\
-0.018 & 0.576 & -0.118 \\
-0.009 & -0.118 & 0.029
\end{array}. \quad (6)
$$

Similarly, the estimated values for the 650-km-radius IMIC, given the outer IC is accounted for using Stephenson et al.'s model[34], are:

$$
\begin{array}{l}
\varepsilon \\
\sigma = \\
\gamma
\end{array}
\begin{array}{l}
0.5 \\
-15.6 \text{ and } C = \\
3.4
\end{array}
\begin{array}{ccc}
0.036 & -0.018 & -0.009 \\
-0.018 & 0.576 & -0.118 \\
-0.009 & -0.118 & 0.029
\end{array}. \quad (7)
$$

In Eqs. 6 and 7, the parameter $\sigma$ in both cases has large negative values indicating that the slow directions deviate from the equatorial plane.

## Data availability

We use Obspy and obspyDMT packages[67,68] to retrieve and process waveform data in this study. Waveforms data and related metadata were accessed through the following data centers, IRIS Data Management Center (http://service.iris.edu), ETHZ (http://eida.ethz.ch/fdsnws), INGV (http://webservices.ingv.it), ORFEUS Data Center (http://www.orfeus-eu.org) and GEOFON Program, GFZ (http://geofon.gfz-potsdam.de).

## Code availability

The codes used in this study to construct global stacks of the direct wavefield, measuring residual travel times between pairs of PKIKP multiples and modeling IC cylindrical anisotropy can be accessed publicly at https://doi.org/10.5281/zenodo.7317680.

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

## Acknowledgements
The authors are grateful to V. Cormier for helping reproduce Fig. S3 from ref. [24] This research was undertaken with the assistance of resources and services from the National Computational Infrastructure (NCI), supported by the Australian Government.

## Author contributions
T.-S. P. and H. T. conceptualized this study and contributed to the interpretation, manuscript writing, and development. T.-S. P. carried out the data processing and analysis and produced the figures.

## Competing interests
The authors declare no competing interests.
