## [Peer Review File · Nature Communications]

REVIEWER COMMENTS

Reviewer #1 (Remarks to the Author):

Summary

The article explores the identification and a good usage of an intriguing set of PKIKP reverberations that sample the IC (inner core) multiple times approximately along its diameter. The deeper interior of the IC has attracted attention in the past two decades with a range of radii predicted for an innermost distinctive sphere. The exotic phases resolved by the authors provide a significant probing mechanism to investigate the central region of IC. They show a clear process of identifying these phases on global stacks via careful selection of event-station configurations. Then travel time residuals between pairs of these exotic reverberations are used to invert for an IC anisotropic model with noticeable weak anisotropy in the identified IMIC (innermost inner core). This introduces a new seismic approach for exploring inner core using a set of phases not used directly before. I recommend this article for publication after authors address some questions and suggested revisions to improve robustness of their presentation of the data and methods sections, some points in discussion, along with a few minor error corrections. The authors' modeling approach for anisotropy in the bulk IC and IMIC is straightforward and robust. I would like to see the presentation of their approach in this paper more cleared up in their revisions.

Questions/Suggestions to the Authors:

Lines 247 – 249 : "Note that a hemispherical structure of the IMIC has recently been suggested, but such a structure cannot be resolved with our dataset that probes near the planetary center." Can the authors explain a bit more in detail how previous predictions of hemispherical differences in the IC anisotropy may or may not be related to their result? Can there be possible tradeoffs? The phases considered in the time residual measurements seem to sample either side of the IC sphere.

Lines 291 – 293 : "After correcting for the instrumental response, seismic records are later bandpass filtered between 10–100 seconds (zero phases, three corners), then grouped in 1-degree distance bins." Typo : word later seems redundant. Edit as "After correcting for the instrumental response, seismic records are bandpass filtered between ..."

Line 298 : Is there a particular reason for the threshold limit to be 5 times the median? Did the authors perhaps make this decision based on the distribution of max amplitudes observed per earthquake?

Figure 5: The captioning of figure 5 seems to have a mix up in identifying the panels. From Lines 322 -328 it looks like 5-A,C,E must refer to aligned traces, stack and spectrogram of PKIKP2 while 5-B,D,F refer to those of PKIKP4. You might need to fix that either in the figure caption or in the text.

Figure S5: Fix the panel reference errors, similar to Fig 5.

Line 329: Is the predicted travel time difference in Eq 3, a ray theoretical prediction? Can the authors justify comparing that with the observed waveform travel time differences computed in a 7 – 13 s band?

Line 341: "In Equation 4, $t_{\text{PKIKP4}} - t_{\text{PKIKP2}}$ are the theoretical or observed differential travel times". Doesn't the 'theoretical' dtt refer to theoretical 'prediction'? It is a bit confusing that the terms theoretical and observed are implied to be synonymous in this sentence. Perhaps clarify what the authors mean here.

Figure S1: Why does this figure contain global stacks pertaining to some events not shown in the location maps of Figure S4? That is if Figure S4 panel titles refer to event origin dates. I suggest the authors show (in supplementary Figure S1 perhaps) the global stacks pertaining to the 16 events used in the study, with phase labels. That would help illustrate

the visual recognition of the said exotic phases clearly in all used events.

Figure S4: For completeness, add a legend or some mentioning to what red and blue symbols mean. Also what do authors' mean by '16 high-events'? The caption of this figure I think can be made a more explanatory.

Figure 1 : Is the delta Xi (uncertainty of sampling direction?) that is marked on panels B & C, quantitatively correlated to the standard deviation of cosine square of xi calculated according to lines 363 – 370?

Lines 365 – 367 : "For mathematical convenience, later used in Equation 1, we calculate the mean and standard deviation of " Should this line be re-written to imply that authors calculate delta cosine square of Xi, instead of delta Xi, since it is mathematically more convenient to be used with Equation 1 ?

Is there a mentioning of how many traces were stacked in each direction before the time residual measurement? I suggest the authors mention information about the the number of traces(stations) used for stacking in each event, which would help validate the uniformity of all data points shown in Figure 3.

Reviewer #2 (Remarks to the Author):

This manuscript presents a nice observation of multiples of PKIKP (with n up to 5) and analysis of the differential times of PKIKP4-PKIKP2 as well as PKIKP3-PKIKP1, which are used the authors to constrain velocity and anisotropy structure of the inner most inner core (IMIC). The PKIKPn observation is quite sound with binned stacking and the travel time data is well analysed. Overall, the paper is well written, and the proposed model of IMIC is important for understanding the evolution history of the Earth's core. I would recommend publication of the paper in NC after some revisions. Here are some issues:

(1) To demonstrate robustness of the observation, the same stacking procedures can be applied to synthetic seismograms with 1D Earth model containing different IMIC. This is the benchmark part of the study.

(2) A frequency band 7-13 s is used. How about other frequency bands?

(3). Measurement of slowness for PKIKP3 would confirm the ray path while the near constant arrivals of PKIKP2 and PKIKP4 at near podal distances imply small slowness.

(4) Although PKIKP5 is observed, it is not used in constraining the IMIC. Thus, the title might be revised a little bit.

(5) in caption of Figure S5. The PKIKP2 and PKIKP4 seem to be typo for PKIKP1 and PKIKP3. And in the last line, the PKIKP1 seems to be for PKIKP3.

Reviewer #3 (Remarks to the Author):

The manuscript presents an interesting study of using newly observed reverberating P wave core phases excited by large earthquakes to investigate the inner core anisotropy. The study convincingly demonstrated these weak phases can be revealed when data from global seismic stations are stacked at the podal and antipodal locations where the focusing effect is particularly strong. The result derived from dense arrays in North America and Europe shows that P wave core phases propagating ~ 50 degrees off from the earth's rotational axis are consistently slower. Based on this result, the authors infer the anisotropic structure of the innermost inner core.

While I find the study quite interesting, I am not convinced that the new result significantly improves our understanding of the inner core structure. Here I list my concerns.

1. The authors argue that the newly observed reverberating core phases complement and

improve currently available information. However, in contrast to previous studies based on the direct PKP phases, these reverberating phases can only be observed when a large dense array is presented in the podal and antipodal locations of a large earthquake. That basically limited the measurements to be along three general ray paths (i.e. straight down below Alaska, the contiguous US, and Europe). It is unclear if the result from only the three ray paths can significantly improve what other measurements have constrained already. No discussion has been given on why the new measurements are expected to outperform the previous measurements and why the new inner core model should be more accurate than the previous models. What is the chance that the mantle and crust structure along these three ray paths are not fully accounted for and introduce systematic travel time biases? Does it make sense to jointly invert the new measurements with the existing measurements? While it could be argued that with more seismic arrays being deployed, more reverberating paths will become available. Realistically though, it will be extremely hard to further improve the ray path distribution considering the massive ocean area and the limited large earthquake locations.

2. The reverberating phases observed are above 10 sec period. Considering the extremely long ray path, I would expect the finite frequency effect can play a significant role when considering the measurement sensitivity. Are the measurements only sensitive to the innermost inner core or the sensitivity kernels are quite broad, and the measurements are also sensitive to the outer inner core? The sensitivity of the measurements is critically important to evaluate the claim in the paper regarding the innermost inner core. I would assume the sensitivity kernel is quite broad not only because the finite frequency effect but also because stations in rather big areas contributed to the stacked waveforms (i.e. a 40 and 20 degree area for podal and antipodal measurements, respectively). It seems the sensitivity kernels of the relative travel time measurements can be further complicated by the fact that each reverberating phase measured by each station can have a rather different ray path. It is also unclear if finite frequency sensitivity has been considered when correct for mantle heterogeneity and if the mantle velocity models used are accurate enough and do not cause any systematic biases.

3. This is more of a style comment. There are many brief discussions related to coda interferometry scattered all over the manuscript. The readers might find some of the sentences out of place. As the authors have published many coda interferometry papers, these discussions might be natural to them but they might not be natural for the readers. As this paper doesn't involve coda interferometry, I feel the authors should limit the discussion on coda interferometry and consider consolidating the discussion into one single section instead of spreading the discussion throughout.

4. Line 185, it is unclear what are dependent and independent parameters.

5. In Figure 3, it would be nice if the uncertainties of the sample angles are plotted in A and B. The uncertainties of the dv/v measurements presented seems to be extremely small. The fact the differences between measurements with similar angles seem to be much bigger than the uncertainties suggest the uncertainty is underestimated. What is the percentage of the stations being removed in the bootstrapping process when estimating uncertainties?

6. It is unclear to me from the discussion if the new measurements can be used to distinguish whether the Bulk IC or IMIC model is more accurate. The fit looks identical for the two models in Figure 3. But the authors seem to prefer the IMIC model in the conclusion (line 271-273).

7. The authors listed the advantages of using the reverberating phases in the first paragraph of the discussion section. It seems some discussion on the potential shortcomings (e.g. limited ray path, broad sensitivity, etc.) is also warranted.

8. Line 231-233, I do not understand the argument regarding the minimal impact from slabs.

9. In Figure 4, please clarify the meaning of orange and blue ray paths.

10. Line 297, what is the N (i.e. width of the time window) used here?

11. Figure 5, the order of A-F discussed in the figure caption is not consistent with what is shown in the plot. Is the result shown in A only corrected for the Earth's ellipticity or also for mantle heterogeneity?

12. Line 332-333, the mean and standard variation are shown on the top instead of the bottom left corners.

13. Figure 6, what is the time window used for the cross-correlation? Should the correlation coefficient of the waveforms be used to evaluate the measurement quality somehow?

Dear Reviewers,

Thank you for your reviewing our manuscript. Your comments have significantly improved the manuscript and strengthened our conclusions.

Please find responses to all points raised in our manuscript. For your convenience, we attach the change-tracked main manuscript and supplementary information at the end of this document. Reference to lines number in the responses corresponds to line numbers in the tracked files.

Best regards,

The Authors

Reviewer #1 (Remarks to the Author):

Summary

The article explores the identification and a good usage of an intriguing set of PKIKP reverberations that sample the IC (inner core) multiple times approximately along its diameter. The deeper interior of the IC has attracted attention in the past two decades with a range of radii predicted for an innermost distinctive sphere. The exotic phases resolved by the authors provide a significant probing mechanism to investigate the central region of IC. They show a clear process of identifying these phases on global stacks via careful selection of event-station configurations. Then travel time residuals between pairs of these exotic reverberations are used to invert for an IC anisotropic model with noticeable weak anisotropy in the identified IMIC (innermost inner core). This introduces a new seismic approach for exploring inner core using a set of phases not used directly before. I recommend this article for publication after authors address some questions and suggested revisions to improve robustness of their presentation of the data and methods sections, some points in discussion, along with a few minor error corrections. The authors' modeling approach for anisotropy in the bulk IC and IMIC is straightforward and robust. I would like to see the presentation of their approach in this paper more cleared up in their revisions.

Thank you for your constructive review. We agree and try to convey that this new set of observations is not only intriguing, given that it has not been reported before, but it also provides a new tool to probe the centremost region of the Earth's inner core. We attempt to further clarify the methods in the revised manuscript addressing your and other reviewers' comments.

Questions/Suggestions to the Authors:

1. Lines 247 – 249 : “Note that a hemispherical structure of the IMIC has recently been suggested, but such a structure cannot be resolved with our dataset that probes near the planetary center.” Can the authors explain a bit more in detail how previous predictions of hemispherical differences in the IC anisotropy may or may not be related to their result? Can there be possible tradeoffs? The phases considered in the time residual measurements seem to sample either side of the IC sphere.

Here, we referred to the most recent results on the innermost inner core (IMIC), which are somewhat controversial. Namely, Brett et al. (2022) and Frost et al. (2021) suggest

hemispherical structures in the IMIC with the opposite polarity. In particular, Brett et al. (2022) reported a 3D tomographic image of the IC using a compilation of several PKIKP datasets. In their model, an IMIC with a slow direction at $\sim 55 \pm 16^\circ$ is confined to the eastern hemisphere. We find similar anisotropic parameters, but we use a spherically symmetric IMIC because our data, sampling near the IC center, is subjected to averaging both hemispheres. We have revised the text to add more details. Lines 289–293.

2. Lines 291 – 293 : “After correcting for the instrumental response, seismic records are later bandpass filtered between 10–100 seconds (zero phases, three corners), then grouped in 1-degree distance bins.” Typo : word later seems redundant. Edit as “After correcting for the instrumental response, seismic records are bandpass filtered between ...”

Edited. Thank you. Line 340.

3. Line 298 : Is there a particular reason for the threshold limit to be 5 times the median? Did the authors perhaps make this decision based on the distribution of max amplitudes observed per earthquake?

A factor of five was chosen empirically. When performing a visual inspection of rejected waveforms of a few events, we found they have apparent anomalies, such as instrumental glitches or artifacts due to data gaps. Lines 347–349.

4. Figure 5: The captioning of figure 5 seems to have a mix up in identifying the panels. From Lines 322 -328 it looks like 5-A,C,E must refer to aligned traces, stack, and spectrogram of PKIKP2 while 5-B,D,F refer to those of PKIKP4. You might need to fix that either in the figure caption or in the text.

Thank you. The references to the labels are now correct. Line 361–364.

5. Figure S5: Fix the panel reference errors, similar to Fig 5.

Done.

6. Line 329: Is the predicted travel time difference in Eq 3, a ray theoretical prediction? Can the authors justify comparing that with the observed waveform travel time differences computed in a 7 – 13 s band?

We used the differential travel time prediction via the “taup” tool (Buland & Chapman, 1983), which is based on ray theory. Via a response to the suggestion made by Reviewer #2, we confirm that the differential travel times of band-limited waveforms agree with the ray theory calculation in Figures S11B and S12B.

7. Line 341: “In Equation 4, $t_{PKIKP4} - t_{PKIKP2}$ are the theoretical or observed differential travel times”. Doesn’t the ‘theoretical’ dtt refer to theoretical ‘prediction’? It is a bit confusing that the terms theoretical and observed are implied to be synonymous in this sentence. Perhaps clarify what the authors mean here.

Thank you. You are right that the denominator is a ray-theoretically predicted quantity. Besides elaborating on the quantity, we also fixed an error in this equation. The observed differential travel time is normalized by the predicted times the multiple PKIKP spend only in the IC, now revised as $\tau_{PKIKP(n+2)}$ and $\tau_{PKIKP(n)}$, to differentiate with $t_{PKIKP(n+2)}$ and $t_{PKIKP(n)}$, which represent the travel times of the entire PKIKP multiples. As mentioned in the text, the relative P-wave speed deviation regarding the exotic multiples defined in Equation 4 is equivalent to similar quantities in many studies using single-passage PKIKP waves. Lines 388–392.

8. Figure S1: Why does this figure contain global stacks pertaining to some events not shown in the location maps of Figure S4? That is if Figure S4 panel titles refer to event origin dates. I suggest the authors show (in supplementary Figure S1 perhaps) the global stacks pertaining to the 16 events used in the study, with phase labels. That would help illustrate the visual recognition of the said exotic phases clearly in all used events.

Thanks for this suggestion. Some events in Figure S1 did not produce good individual observations of PKIKP exotic pairs for measuring differential travel times in the podal or antipodal distance ranges. Therefore, they were not shown in Figure S4. However, thanks to your suggestion, we now supplement figures similar to Figure 5 and Figure S5 for all 16 events producing the differential time measurements.

9. Figure S4: For completeness, add a legend or some mentioning to what red and blue symbols mean. Also what do authors' mean by '16 high-events'? The caption of this figure I think can be made a more explanatory.

We meant the 16 events that produced sufficiently high-quality waveforms to measure the differential travel time of PKIKP multiple pairs. The caption has been revised, and we have added a description for the red and blue symbols. (See new caption of Figure S4.)

10. Figure 1 : Is the $\Delta \xi$ (uncertainty of sampling direction?) that is marked on panels B & C, quantitatively correlated to the standard deviation of cosine square of ξ calculated according to lines 363 – 370?

As mentioned in the Methods section, we estimate uncertainty for $\cos^2 \xi'$ due to its mathematical convenience in Equation 1. The $\Delta \xi'$ shown in Figure 1 is an equivalent quantity after converting from this counterpart, $\Delta \cos^2 \xi'$ (please see the new Figure 3A with the newly added error bars, $\Delta \xi'$).

11. Lines 365 – 367 : “For mathematical convenience, later used in Equation 1, we calculate the mean and standard deviation of ” Should this line be re-written to imply that authors calculate delta cosine square of ξ , instead of delta ξ , since it is mathematically more convenient to be used with Equation 1?

Thanks, you are right. This sentence has been revised to highlight our preference for $\cos^2 \xi'$ rather than ξ' due to the mathematical convenience. Lines 415–419.

12. Is there a mentioning of how many traces were stacked in each direction before the time residual measurement? I suggest the authors mention information about the number of traces(stations) used for stacking in each event, which would help validate the uniformity of all data points shown in Figure 3.

We provided the statistics in Figure S13C in the supporting information.

Reviewer #2 (Remarks to the Author):

This manuscript presents a nice observation of multiples of PKIKP (with n up to 5) and analysis of the differential times of PKIKP4-PKIKP2 as well as PKIKP3-PKIKP1, which are used by the authors to constrain velocity and anisotropy structure of the innermost inner core (IMIC). The PKIKPn observation is quite sound with binned stacking and the travel time data is well analysed. Overall, the paper is well written, and the proposed model of IMIC is important for understanding the evolution history of the Earth's core. I would recommend publication of the paper in NC after some revisions. Here are some issues:

Thank you for your comments, which help strengthen our manuscript. The suggestion of a numerical benchmarking experiment is particularly beneficial, and we used it to address the point about the measurement sensitivity raised by another reviewer.

(1) To demonstrate robustness of the observation, the same stacking procedures can be applied to synthetic seismograms with 1D Earth model containing different IMIC. This is the benchmark part of the study.

Thank you for the constructive comments. To benchmark our method, we have performed a series of full waveform simulations, whose details are presented in Text S1 in the Supporting Information. The finite frequency results are also beneficial in establishing the feasibility of using the PKIKP multiples to probe the ~650-km radius IMIC.

(2) A frequency band 7-13 s is used. How about other frequency bands?

When observing the PKIKP multiples at regional arrays, we endeavored to find the shortest period bands possible to resolve the IMIC. After many trials, the 7–13 s band was empirically the shortest period band that can be applied to retain 16 differential travel time measurements. This is mentioned in lines 169–175.

(3). Measurement of slowness for PKIKP3 would confirm the ray path while the near constant arrivals of PKIKP2 and PKIKP4 at near podal distances imply small slowness.

We think our implied ray paths (as shown in Figure 1C) for the PKIKP3 observations and synthetics (e.g., in Figure S5) can be corrected without requiring further precision of their

slowness. As can be seen in Figures S5 and S12, the observed and synthesized PKIKP3 arrivals are well aligned against ray-theoretical predictions.

(4) Although PKIKP5 is observed, it is not used in constraining the IMIC. Thus, the title might be revised a little bit.

We acknowledge your suggestion. However, we think that our unprecedented observation of PKIKP5 waves in the direct seismic wavefield is as exciting as the implication on the IMIC using the new sets of differential travel time measurements. Hopefully, this justifies our intention to retain the title as is.

(5) in caption of Figure S5. The PKIKP2 and PKIKP4 seem to be typo for PKIKP1 and PKIKP3. And in the last line, the PKIKP1 seems to be for PKIKP3.

Thank you, this was indeed our omission. Another reviewer also raised it. It has been fixed.

Reviewer #3 (Remarks to the Author):

The manuscript presents an interesting study of using newly observed reverberating P wave core phases excited by large earthquakes to investigate the inner core anisotropy. The study convincingly demonstrated these weak phases can be revealed when data from global seismic stations are stacked at the podal and antipodal locations where the focusing effect is particularly strong. The result derived from dense arrays in North America and Europe shows that P wave core phases propagating ~ 50 degrees off from the earth's rotational axis are consistently slower. Based on this result, the authors infer the anisotropic structure of the innermost inner core.

While I find the study quite interesting, I am not convinced that the new result significantly improves our understanding of the inner core structure. Here I list my concerns.

We appreciate your comments that our study is interesting, and we use the opportunity to turn your constructive criticism into improvements of our manuscript, in particular, to emphasize better how this new dataset differs from the previous studies and why it improves our understanding of the inner core.

1. The authors argue that the newly observed reverberating core phases complement and improve currently available information. However, in contrast to previous studies based on the direct PKP phases, these reverberating phases can only be observed when a large dense array is presented in the podal and antipodal locations of a large earthquake. That basically limited the measurements to be along three general ray paths (i.e. straight down below Alaska, the contiguous US, and Europe). It is unclear if the result from only the three ray paths can significantly improve what other measurements have constrained already. No discussion has been given on why the new measurements are expected to outperform the previous measurements and why the new inner core model should be more accurate than the previous models. What is the chance that the mantle and crust structure along these three ray paths are not fully accounted for and introduce systematic travel time biases? Does it make sense to jointly invert the new measurements with the existing measurements? While it could be argued that with more seismic arrays being deployed, more reverberating paths will become available. Realistically though, it will be extremely hard to further improve the ray path distribution considering the massive ocean area and the limited large earthquake locations.

Thank you for the thoughtful comments. Below we respond to your questions in three groups.

It is unclear if the result from only the three ray paths can significantly improve what other measurements have constrained already. No discussion has been given on why the new measurements are expected to outperform the previous measurements.

Due to the familiar event and station locations, it could be misunderstood that the sampling volumes of the deep Earth are the same as in the previous studies. However, because we now use the *podal* paths connecting the sources with the nearly antipodal points and back to the nearby receivers, the new paths through the Earth's inner core are unique and different from most paths used before. Let us take an example of an excellent observation of PKIKP4 from the Anchorage event (Southern Alaska), which samples the center of the IC. This sampling direction is unique. In theory, it would only be possible to achieve a direct PKIKP wave if an earthquake occurred near the antipode of Anchorage, which is below the southern tip of South Africa. This location is in the middle of an oceanic plate and thus is void of large earthquakes. Not less significant is an additional novelty. Although we used three SSI events (Figure S4), the residual travel times of their PKIKP3-PKIKP pairs have substantially reduced likely strong slab effects from both sides at both source and receivers due to their ray path proximity (Figure S9).

In conclusion, the exotic PKIKP multiples bring fresh insights into the IC studies even though the current distributions of earthquakes and seismograph networks are used. The new observations take the existing source-receiver configuration to a new level.

You correctly point out that the new observations are tied to regional, dense seismic networks. We use the sampling along the north-south direction (a.k.a. the polar direction), which was previously extensively sampled along the anomalous South-Sandwich Islands-Alaska paths, to demonstrate our arguments. We have revised the first two paragraphs in the Discussion (Lines 234–263) to clarify the advantages of the new observations.

Why the new inner core model should be more accurate than the previous models? Does it make sense to jointly invert the new measurements with the existing measurements?

As the mathematician G. Box famously pointed out, all models are wrong, but some are useful. Perhaps it has not been clearly conveyed, but we do not argue that our inferred IC models have higher accuracy than the previously proposed models. In Figure 3, we interpret this study's findings in the context of the most recent results of our research group (Costa de Lima et al., 2022; Stephenson et al., 2021; Tkalčić et al., 2022) (Lines 274–278) and the work of Brett et

al. (2022) (Lines 289–293). All recent attempts using independent data converge in the sense that they infer the existence of an anisotropically distinct IMIC. Perhaps a future effort to jointly invert all possible datasets (including free oscillations) for the 3D structure of the IC would be feasible, likely using a similar method to Brett et al. (2022).

What is the chance that the mantle and crust structure along these three ray paths are not fully accounted for and introduce systematic travel time biases?

Indeed, some influence of heterogeneities from upper layers on our travel time measurements is inevitable, like in all geophysical studies, where previous and more established knowledge is used. The rationale in all state-of-the-art inner core studies is that we must rely on the mantle models, given that they are better sampled by the seismic data and better documented than the inner core. We account for mantle heterogeneity using several representative mantle P-wave models (Lines 205–209). To avoid the influence of a particular mantle model, we repeated the analysis for four P-wave mantle models. As the text mentions, the general IC anisotropic trend in Figures 3, S6–58, is preserved. We hope that the rigorous practice in treating mantle heterogeneities mitigates the chance of our travel time measurements being systematically biased.

2. The reverberating phases observed are above 10 sec period. Considering the extremely long ray path, I would expect the finite frequency effect can play a significant role when considering the measurement sensitivity. Are the measurements only sensitive to the innermost inner core or the sensitivity kernels are quite broad, and the measurements are also sensitive to the outer inner core? The sensitivity of the measurements is critically important to evaluate the claim in the paper regarding the innermost inner core. I would assume the sensitivity kernel is quite broad not only because the finite frequency effect but also because stations in rather big areas contributed to the stacked waveforms (i.e. a 40 and 20 degree area for podal and antipodal measurements, respectively). It seems the sensitivity kernels of the relative travel time measurements can be further complicated by the fact that each reverberating phase measured by each station can have a rather different ray path.

Thanks for the thoughtful comment. To respond to the point raised here, we conducted a new series of numerical experiments, which also helped to respond to another reviewer. The details

of the experiments are presented in Text S1 of the supplementary information. Here, we summarise the main arguments relevant to the point you raised,

- The benchmark experiment demonstrates that the relatively long-period signals are exclusively sensitive in sampling a 650-km-in-radius IMIC.
- The sensitivity of the measurement method is benchmarked for measurements performed on the synthetic waveforms filtered in the same broadband as the data. Even though the measurement sensitivity of PKIKP4-PKIKP2 could degrade beyond 30° epicentral distances, we only have two (out of 16) measurements in these ranges, which are in the region where the potential errors are sufficiently small.

In addition, we found an error in reported epicentral distance ranges for the podal and antipodal settings: $>155^\circ$ instead of $>160^\circ$ for the PKIKP3-PKIKP pair and $<50^\circ$ instead of $<40^\circ$ for the PKIKP4-PKIKP2 pair. The numerical experiments (Text S1) have confirmed the measurement sensitivity of the pairs for these expanded distance ranges, as mentioned above.

It is also unclear if finite frequency sensitivity has been considered when correct for mantle heterogeneity and if the mantle velocity models used are accurate enough and do not cause any systematic biases.

The finite frequency effects in correcting for mantle heterogeneities is not an issue because the possible discrepancy between corrections using finite-frequency and ray-theoretical predictions would be almost constant regardless of sampling directions. Thus, it might only result in a possible baseline shift for all measurements, which is already accounted for by the γ parameter in Equation 1.

3. This is more of a style comment. There are many brief discussions related to coda interferometry scattered all over the manuscript. The readers might find some of the sentences out of place. As the authors have published many coda interferometry papers, these discussions might be natural to them but they might not be natural for the readers. As this paper doesn't involve coda interferometry, I feel the authors should limit the discussion on coda interferometry and consider consolidating the discussion into one single section instead of spreading the discussion throughout.

Thanks for pointing this out; we agree. We would like to make it clear from the manuscript that the observations of the exotic PKIKP multiples were inspired by our previous and ongoing work on the coda correlation wavefield. However, we identified some sentences referring to the correlation wavefield that is ‘out of place’. Consequently, we have revised the manuscript to minimize the references and restrict extensive discussion related to the correlation wavefield. This is now limited to two paragraphs, one in the introduction and another in the discussion section. Lines 138–139, 145–147, 148–156.

4. Line 185, it is unclear what are dependent and independent parameters.

In Equation 1, $\cos^2 \xi'$ is the explanatory (a.k.a. independent) variable and $\Delta v/v$ is the response (a.k.a. dependent) variable. We have revised the text (Lines 197–199 and 427) to clarify the confusion.

5. In Figure 3, it would be nice if the uncertainties of the sample angles are plotted in A and B. The uncertainties of the dv/v measurements presented seems to be extremely small. The fact the differences between measurements with similar angles seem to be much bigger than the uncertainties suggest the uncertainty is underestimated. What is the percentage of the stations being removed in the bootstrapping process when estimating uncertainties?

We have added the error bar for the angle ξ' in Figure 3. In the bootstrapping process, we randomly resampled the original waveform set with replacement and did not control the number of waveforms to be removed or repeated in a re-sample.

6. It is unclear to me from the discussion if the new measurements can be used to distinguish whether the Bulk IC or IMIC model is more accurate. The fit looks identical for the two models in Figure 3. But the authors seem to prefer the IMIC model in the conclusion (line 271-273).

This is a valid point; we presented our thoughts in the original manuscript in the paragraph just before the discussion section (Lines 210–218). Indeed, the differential travel time data fitting itself cannot distinguish between the two as most sampling paths are near the Earth’s center. However, the new data confirm the existence of IMIC because the OIC has been broadly studied with differential travel time measurements, and no such anisotropy results have been obtained. Thus, a distinct inner shell is required to explain the tilted slow axis of propagation.

7. The authors listed the advantages of using the reverberating phases in the first paragraph of the discussion section. It seems some discussion on the potential shortcomings (e.g. limited ray path, broad sensitivity, etc.) is also warranted.

Thanks. We have added a new paragraph in the Discussion on the limitations with thoughts on the way forward. Lines 265–273.

8. Line 231-233, I do not understand the argument regarding the minimal impact from slabs.

Thank you. As this point was not evident in the original manuscript, in the revision (Line2 234–246), we explain that for the PKIKP3-PKIKP pair of an SSI earthquake recorded by the Alaskan array, due to their ray path proximity (Figure 1C), the effect of slabs on both sides will be reduced substantially.

9. In Figure 4, please clarify the meaning of orange and blue ray paths.

In panel C, blue and orange lines are ray paths of PKIKP2 and PKIKP4 arrivals. In panel D, blue lines are the ray path of the correlated I2* features resulting from highly multiple I-waves plotted in orange. The figure caption has been modified accordingly. Lines 308–313.

10. Line 297, what is the N (i.e. width of the time window) used here?

$N = 72,000$ points corresponding to 120-minute-long seismograms having 10 samples per second. Line 336.

11. Figure 5, the order of A-F discussed in the figure caption is not consistent with what is shown in the plot. Is the result shown in A only corrected for the Earth's ellipticity or also for mantle heterogeneity?

Thank you. We have fixed the label order.

In Figures 5 and S5, ray-theoretical predictions are corrected for the Earth's ellipticity and mantle heterogeneity using the DETOX-P3 mantle model, which was mentioned in the original main text. We have added it to the captions of Figures 5 and S5.

12. Line 332-333, the mean and standard variation are shown on the top instead of the bottom left corners.

Fixed. Lines 380–381.

13. Figure 6, what is the time window used for the cross-correlation? Should the correlation coefficient of the waveforms be used to evaluate the measurement quality somehow?

We generally use around 120 seconds surrounding the predicted arrivals to measure differential travel times. The exact windows used for the correlation measurements have been provided in a new set of supplementary figures. Due to the relatively small number of earthquakes to be examined, we manually inspect the measurement quality with cross-correlation methods. The correlation coefficient would be helpful in a more extensive search for PKIKP multiples in future studies.

**Observations of up-to-fivefold reverberating waves through the Earth's center:**
**distinctly anisotropic innermost inner core**

Thanh-Son Pham^{1,*} and Hrvoje Tkalčić¹

¹Research School of Earth Sciences, The Australian National University, Canberra ACT
Australia

*Corresponding author: [T.-S. Pham \(thanhson.pham@anu.edu.au\)](mailto:Thanhson.pham@anu.edu.au)

**Abstract**

[revised manuscript text omitted]

by the pronounced expression of PKIKP multiple counterparts in the correlation wavefield²⁹
utilizing coda records several hours after the origin time.

There are features in the global coda correlograms that looked similar to PKIKP multiples and
were explained to arise due to the similarity of seismic phases late, weak arrivals after large
earthquakes rather than "reconstructed" body waves²⁹. The emergence of the core-sensitive
signals in the coda-correlation wavefield thus inspired us to search for exotic reverberations in
the direct seismic wavefield that results in the correlation features' formation. Unlike the exotic
correlation features, whose complex geometrical sensitivity kernels to Earth's internal
structures must yet be fully understood⁴⁶, the observed multiple PKIKP waves are practical

Commented [SP2]: Revision to reduce 'scattered' reference of coda interferometry. Response to Reviewer 3, point #3.

because their sensitivity can be mapped along their ray paths. Here, we applied these new
observations to constrain the IMIC.

Commented [SP3]: Revision to reduce technical reference to coda interferometric concepts. Response to Reviewer 3, point #3.

**Figure 2. Global stack for the 22 Jan 2017, Mw=7.9 Solomon Islands earthquake. (A)**

**Histogram of seismic waveforms as a function of epicentral distance in 1-degree bins. (B) and**

(C) Global stacks spanning 0–55 minutes and 55–110 minutes. Exotic podal and antipodal
reverberations up to five multiples (with near-horizontal slope) are labeled by red fonts.

**New Constraint of Distinct Anisotropy in the Innermost Inner Core**

As the podal (receivers near the source) and antipodal (receivers on the antipodal side from the
source) waves spend multiple passages through Earth’s inner core, their travel times can
provide constraints on the IC structures once they are corrected for contributions of source
location errors, the Earth’s ellipticity, and mantle heterogeneities. In the Methods section, we
describe a procedure to measure the travel time residuals for pairs of exotic arrivals, such as

PKIKP4-PKIKP2 and PKIKP3-~~PKIKP4~~, on the stacked waveforms over dense regional
seismic networks. In our regional observations, we can use a slightly shorter period band (i.e.,
7–13 s in Figures 5 and S5) than in the global stacks (i.e., 10–100 s in Figures 2 and S1), as the
propagation paths to array elements are likely to experience fewer heterogeneities in the
mantle. Occasionally, even shorter period observations could be made (e.g., 1–10 s for the 2018
Anchorage earthquake). However, the 7–13 s period band was found to be suitable for
obtaining as many as 16 differential travel time measurements of the PKIKP multiple pairs (see
Figures 6 and S4). Thus, we proceed with this period band in the subsequent analysis.
Supplementary Text S1 presents finite-frequency numerical experiments using the spectral
element method⁴⁷ in a 2D Earth section to demonstrate the feasibility and measurement
sensitivity of the 7–13 s PKIKP multiples in probing a 650-km-radius IMIC.

We consider the cylindrically anisotropic model of the Earth’s inner core to fit the residual
travel time residuals relative to the ak135 reference model⁴⁸. In a cylindrical model, the
relatively small perturbation from the background velocity of the Earth’s IC is expressed as a
function of a single dependent parameter, the sampling angle, ξ' ; specifically⁴⁹,

Commented [SP4]: We change all PKIKP1’s to PKIKP to homogenise the phase naming convention.

Commented [SP5]: Response to Reviewer 2, point #2 on the choice of 7–13 period band.

Commented [SP6]: Here we refer to the new Text S1 in the supplementary information describing a series of numerical experiments on the observation robustness and measurement sensitivity. Response to Reviewer 2, point #1 and Reviewer 3, point #2.

$$\frac{\Delta v}{v} = \gamma + \varepsilon \cos^2 \xi' + \sigma \sin^2 \xi' \cos^2 \xi'. \quad (1)$$

In Equation 1, ε and σ are the controlling parameters of the model, and the baseline shift γ
 accounts for the uncertainty of the 1D reference model. Equation 1 is thus a quadratic function
 of $\cos^2 \xi'$. In this equation, a representative sampling angle, ξ' , is defined to represent the
 sampling direction of the exotic arrival pairs relative to the Earth's rotation axis (see more
 details in the Methods section). We estimate the parameters for two anisotropic models of the
 Earth's IC, including (i) the bulk IC model assuming the directional dependence of seismic
 wave speed does not change with depths and (ii) the model with an IMIC, consisting of two
 anisotropically different domains, a concentric outer shell, and an innermost ball, i.e., IMIC.
 In the model with an IMIC, the anisotropic strength and depth extent, H , of the outer layer from
 the inner core boundary are fixed to the recent model proposed by ref.³⁴, in which $H = 650$
 193 km, $\varepsilon = 1.45\%$, $\sigma = -1.07\%$, and $\gamma = 0$, as our data cannot independently constrain the outer
 layer's parameters.

In this study, we use the orthogonal distance regression method⁵⁰ to estimate the anisotropic
 parameters (see the Methods section) because this method can account for the measurement
 uncertainties of both ~~dependent explanatory variable (i.e., $\cos^2 \xi'$ eos —)~~ and ~~independent~~
 ~~response parameters~~variable (i.e., $\Delta v/v$) of Equation 1, which are both significant in our
 observed data. The mean anisotropy models are plotted in Figure 3 in dark solid lines. Light
 blue lines represent the models' uncertainty by simulating the parameters with correlated
 uncertainty using the Monte Carlo method. The bulk IC models indicate that P-wave travels
 through the IC at the lowest speed at around $\xi' \approx 48^\circ$, while the speed is $\sim 2.8\%$ faster along
 the polar direction and 1.7% faster along the equatorial plane (Figure 3A). In the two-layer IC
 model, the slow direction in the IMIC remains almost unchanged, but P-waves are $\sim 4.0\%$ and
 3.4% faster when traveling along the polar direction and equatorial plane (Figure 3B). Similar

Commented [SP7]: Clarification of dependent and dependent parameters. Response to Reviewer 3, point #4.

models with the slow direction offset significantly from the ERA are also observed when other
mantle models are used to correct for mantle heterogeneities, i.e., MIT-P08⁵¹ in Figure S6,
LLNL-G3Dv3⁵² in Figure S7, and no mantle correction applied in Figure S8. Thus, we consider
that the changing in patterns of P-wave anisotropy with depths in the IC is robustly observed.

Note that our measurements sample near the Earth's center and have minimal depth sensitivity,
so we cannot directly favor the bulk IC model (Figures 3A and 3B) or the IC model with an
IMIC (Figures 3C and 3D) based on the data fits alone. Instead, the centermost-sensitive
observations demonstrate a prominent anisotropic pattern in the IMIC (Figure 3B) with slow
directions residing at mid-range latitudes. This property, in line with pioneering
observations^{1,33} drives the anisotropic pattern in the bulk IC models (Figure 3A) because the
anisotropy in the outer IC (OIC) is weaker than the IMIC with slow directions around the
equatorial plane² (a representative anisotropic model of OIC from ref.³⁴ is plotted in Figure
3B). This enables us to infer a distinctively anisotropic IMIC from the new datasets.

Commented [SP8]: We have added this sentence to make it clear that, we cannot prefer the bulk IC or the IC with an IMIC model with the probe having little depth sensitivity. Indeed, the original paragraph was to elaborate on that point. Response to Reviewer 3, point #6.

Commented [SP9]: We have added the error bar for ξ' in the left panels. Response to Reviewer 3, point #3.

**Figure 3. Cylindrically anisotropic model of Earth's IC inferred from exotic PKIKP**
 **multiples' travel times.** Fractional velocity (see the Methods section) and fitting curves are
 plotted as a function of ξ' , the representative sampling direction of the ray (left) (see Figure 1),
 and $\cos^2 \xi'$ (right). All measured differential travel times are corrected for mantle
 heterogeneities using the DETOX_P3 model⁵³. Associated uncertainties are plotted by error
 bars. Dark solid lines are the optimal anisotropic models parameterized in Equation 1, and light
 blue opaque lines represent the uncertainty surrounding the optimal model. Various broken
 lines show models from previous studies (see the legend). The top row (A) compares our
 inferred bulk IC model with other models, while the bottom row (B) compares our inferred

IMIC model with other models. Yellow lines in B) show a representative anisotropy model of
the outer inner core (OIC) taken from ref.³⁴, which is used to account for the OIC structure and
obtain our IMIC models.

Discussion

~~Despite much more effort required to obtain the differential travel time measurements between~~
~~pairs of exotic arrivals than the widely used PKP wave differential travel times, this approach~~
~~has clear advantages for sampling the Earth's IC. The direct observations of PKIKP multiples~~
~~using regional seismic arrays equip seismologists with new seismic phases to sample the center~~
~~of the Earth's IC. This new approach has clear advantages even with the existing distribution~~
~~of earthquakes and seismograph networks. Firstly, the observed reverberations provide a~~
~~unique sampling style of the Earth's IC along the north-south direction. Although this direction~~
~~has been sampled with the South Sandwich Islands (SSI) events recorded in Alaska⁵⁴⁻⁵⁷, due~~
~~to the epicentral distance range, only the outer parts of the IC were sampled. However, our~~
~~observation of the exotic phases from the 2018 Anchorage event, recorded by the elements by~~
~~the Alaskan branch of the US Transportable Array (Figure 5) within the 10° epicentral distance~~
~~range now sample the very center of the IC due to their unique podal geometry (see ray paths~~
~~in Figure S9).~~

~~Firstly, uncertainties due to earthquake location errors are mitigated by differentiating the~~
~~arrival times of two exotic arrivals in stacked records. Secondly, the uncertainties due to~~
~~earthquake location errors are mitigated by measuring the differential travel times of a pair of~~
~~exotic phases. The impacts both Alaskan and SSI slabs might have on the travel times are~~
~~mitigated for the measurements associated with the three SSI events in 2018 (see their location~~
~~in Figure S4) thanks to the proximity of PKIKP and PKIKP3 ray paths at both source and~~
~~receiver sides (see Figure 1C). Secondly, w~~When the stations and events are restricted to the

Commented [SP10]: We have revised this paragraph to elaborate on the advantages of the new IC probe. Response to Reviewer 3, point #1.

Commented [SP11]: Revision to clarify the minimal impact of slabs on new IC probe. Response to Reviewer 3, point #8.

Commented [SP12]: Revision to clarify the minimal impact of slabs on new IC probe. Response to Reviewer 3, point #8.

podal and antipodal configurations (i.e., $<540^\circ$ for PKIKP4-PKIKP2 and $>16055^\circ$ for PKIKP3-
PKIKP), the ~~exotic-PKIKP~~ reverberating arrivals sample the centermost 650 km of the IC
several times (see Figures 1B and 1C), which amplifies the evidence for any travel-time
anomalies. ~~Thirdly, the observed reverberations provide complimentary sampling to the~~
~~Earth's IC, particularly along the north-south direction, mainly sampled with the anomalous~~
~~South Sandwich Islands (SSI) to Alaska paths⁵³⁻⁵⁶. The present study's observation of the~~
~~exotic arrivals from the 2018 Anchorage event, recorded by elements by the Alaskan branch~~
~~of the US Transportable Array (Figure 5) within 10° , uniquely complements the SSI-Alaska~~
~~polar paths. It is because the ray path sampling the center of the IC steeply beneath the Alaskan~~
~~peninsula with minimal impact from the SSI and Alaskan slabs along their ray paths.~~

One of the challenges in expanding the use of the exotic PKIKP multiples is the involvement
of large, dense seismic arrays such as USArray or AlpArray. The Earth's heterogeneous
structures beneath large arrays require observations at longer periods, leading to broad
sensitivity to the IC. Due to limited data access, the large-scale ChinArray has not been
explored in this study. Overall, it will be challenging to introduce brand new sampling
directions to the IC unless other large-array projects get underway in the next several decades.
Thus, future attempts in this direction should focus on identifying higher-frequency
observations using smaller aperture networks, e.g., down to national-scale and/or local
seismograph networks, which are more available worldwide.

As shown in Figure 3, this study's main findings of IC's cylindrically anisotropic models with
the slowest direction at $\xi' \approx 48^\circ$ (Figure 3B) are consistent with previous studies concerning
the IMIC, such as the comprehensive absolute PKIKP datasets released by the ~~International~~
~~Seismological Centre~~ISC^{1,34}, dedicated handpicked datasets³⁵⁻³⁷ (Figure 4B), and new
constraints from the global correlation wavefield³² (Figure 4C). Both the *bcc* crystallographic

Commented [SP13]: This paragraph is added to discuss on the limitations of our approach and thoughts on the way forward. Response to Reviewer 3, point #7.

structure of iron²³ and *hcp* iron^{36,58} can have slow directions at oblique angles relative to the
ERA, depending on the orientation of iron crystals, which agrees with our results. Although
the *hcp* iron enables more approachable studies, recent *ab initio* calculations at the IC
temperature and pressure conditions suggest that *bcc* crystals^{59,60} could inherently explain the
reduced shear modulus of the Earth's IC, high anisotropy, high Poisson's ratio, and high
attenuation.

The properties in the innermost 650-km shell of the IC are significantly different from the outer
shell (Figure 3A), characterized by weak anisotropy with the fast axis along the Earth's rotation
axis and a slow direction residing in the equatorial plane^{54,61} (see the schematic representation
of this anisotropic pattern in Figure 4A). Note that a hemispherical structure of the IMIC has

~~recently been suggested by ^{38,62,38}, but such a structure cannot be resolved with our dataset that~~
~~probes near the planetary center, been suggested in two recent studies^{38,62}.~~ When inverting for
the 3D structures of the IC using single-passage PKIKP probes, the former study³⁸ found an
IMIC confined in the eastern hemisphere with a slow direction at $\sim 55 \pm 16^\circ$. Our data sampling
near the planetary center yields a similar value for the slow propagation direction. Several

geodynamical models have been invoked to explain the changes in the anisotropic properties
with depth, including (1) diminishing strength of thermal convection over time⁶³; (2)
preferential crystallization due to the transition in the deformation pattern over time coupled
with density stratification⁶⁴; and (3) the IC's growth could have been conditioned by the
sedimentation of light elements at the ICB, which is linked to chemical variations in the outer
core^{65,66}.

Commented [SP14]: Comment on the relationship between our results and previously published results (Brett et al., 2022 and Frost et al., 2021). Response to Reviewer 1, point #1.

**Figure 4. Schematic model of IC containing IMIC and various body-wave probing**
 **methods to the Earth's center.** (A) On the left panel, the IC model contains IMIC with two
 distinct P-wave transversely isotropic patterns in OIC and IMIC; black and red bars represent
 the fast and slow anisotropic direction; ~~the Earth's rotation axis (ERA) is represented by a~~
 ~~vertical gray line~~ a vertical gray line represents the Earth's rotation axis (ERA). On the right
 panel; cylindrically anisotropic models of OIC (black dashed line) and IMIC (gray line).

Schematic views of IMIC-sampling methods using (B) absolute PKIKP waves, (C) PKIKP
multiples (this study) – blue and orange ray lines represent PKIKP2, and PKIKP4 ray paths,
and (D) correlation feature I_2^* at the two receivers (blue ray paths) results from cross-
correlating high-multiples of PKIKP (orange ray paths). The cross-sections of the Earth in the
right column contain ray path segments sampling the OIC and IMIC. Yellow stars are sources,
and blue inverted triangles are receivers.

Commented [SP15]: Description on the ray path colours and symbols. Response to Reviewer 3, point #9.

In conclusion, we have employed the modern global network to compute stacks of seismic
wavefields induced by individual earthquakes. We report unprecedented robust observations
of podal and antipodal reverberations of compressional waves through the Earth's bulk.
Opportunistically, dense networks at continental scales such as the USArray, including
mainland and Alaska deployments, or the AlpArray in Europe are exploited to sample the
Earth's center by measuring the differential residual between pairs of exotic arrivals. The
inferred model supports the existence of the anisotropically-distinctive IMIC from its outer
shell, which might indicate a fundamental shift in the IC's growth regime in the Earth's past.
We now have enough seismological evidence from several different lines of investigation about
the existence of IMIC. Future efforts should be directed toward characterizing the IMIC-OIC
transition (its depth and nature). The findings reported here are a consequence of the
unprecedentedly growing volume of digital waveform data and will hopefully inspire further
scrutiny of existing seismic records for revealing hidden signals that shed light on the Earth's
deep interior.

**Methods**

**Data retrieval and pre-processing**

Seismic waveforms are gathered from multiple data centers, including IRIS, ORFEUS, ETH,
INGV, and GFZ. Data from four later centers improve coverage in Europe, mainly used in the

detection pair of PKIKP3-PKIKP4 for events in the Kermadec Islands region. We only use
data from stations that continuously record for 120 minutes from the event origin time. All
retrieved seismic waveforms are corrected for instrumental responses to obtain velocity
seismograms and resampled to 10 samples per second (sps) in the preparation stage. Thus $N =$
72,000 (7,200 s \times 10 sps) is the number of points in each waveform. All data management and
processing tasks are performed using the obspyDMT⁶⁷ and ObsPy⁶⁸ packages.

Commented [SP16]: Response to Reviewer 3, point #10.

Construction of global stack

Here, we describe the procedure to construct the global stack of the direct wavefield. After
correcting for the instrumental response, seismic records are ~~later~~ bandpass filtered between
10–100 seconds (zero phases, three corners), then grouped in 1-degree distance bins. Next, we
deployed a median filter to remove seismic traces with anomalous amplitudes due to possible
instrument malfunctions or glitches. Particularly, the median value of maximum absolute
amplitudes for a distance bin is obtained,

Commented [SP17]: Reviewer 1, point #2.

$$v_{med} = med_{k=1,M} \left(\max_{i=1,N} |v_i^k| \right) \quad (2)$$

where M is the number of traces in a bin and $N = 72,000$ is the number of samples in time. Any
waveform trace having its maximum absolute amplitude, $\max_{i=1,N} |v_i^k|$, larger than five times the
median value, v_{med} , is discarded from the further processing. The factor of five was chosen
empirically because anomalous records were efficiently rejected during a visual inspection for
some events. There are no other measurements applied for quality control. The remaining
traces in the bin are summed (i.e., linearly stacked) to render one vertical stripe in the 2D global
stacks (e.g., Figures 2 and S1). Because all waveforms are from a common earthquake, we do
not apply any amplitude normalization, which alters the relative amplitudes of the waveforms.
Instead, to improve the visibility of arrivals at significant lapse times like PKIKP4 and PKIKP5

Commented [SP18]: Response to Reviewer 1, point #3.

in Figure 2, we multiply all binned stacks by a common polynomial of elapsed time, $f(t) =$
 $(t * 10)^4$. We use linear interpolation in visualizing global stacks in Figures 2 and S1.

**Measuring residual travel times of exotic multiples**

Figure 55. Observations of PKIKP2 and PKIKP4 phases in the seismic wavefield from the
 30/11/2018, Mw 7.1 Anchorage earthquake. (A) Seismic records from the Alaskan network
 are aligned with the predictions of PKIKP2 arrivals, corrected by the Earth's ellipticity⁶⁹. The
 waveforms are bandpass filtered in the period band of 7–13 seconds. (CB) Linear stack of
 individual waveforms. (EE) The spectrograms of stacked waveforms before filtering show the
 frequency content variation as a function of time. (BD, DE, F) Similar to A), CB), and EE) but
 for the PKIKP4 arrivals.

Commented [SP19]: Correction for panel mislabels.
 Response to Reviewer 1, point #4; Reviewer 2, point #5; and
 Reviewer 3, point #11.

To measure the residual times for the exotic pair, we initially align individual waveforms
 according to theoretical predictions of PKIKP(n) and PKIKP(n+2) (n = 1, 2) according to the
 *ak135* reference model⁴⁸ and corrected for the Earth's ellipticity⁶⁹ and mantle heterogeneities
 using several P-wave 3D mantle models^{51–53,70}. It is worth noting that in the measuring
 procedure, stacking is necessary because the arrivals on individual waveforms have low signal-
 to-noise ratios, especially for the third and higher multiples. Figure 5 shows data from ~350
 vertical seismograms in the Alaskan branch of the US Array recording the 30 Nov 2018 Mw
 7.1 Anchorage earthquake, where the DETOX-P3 model⁵³ is used to correct for mantle
 heterogeneities. The time corrections are applied to shift the waveforms accordingly (Figures
 5A and 5B), and they are then stacked to enhance signal-to-noise ratios (Figures 5C and 5D).
 The stacked waveforms over the entire array show prominent signals of remarkable similarity
 for the two late arrivals, so the residual of the differential travel time of the arrival pair to the
 reference model can be determined using the cross-correlation:

$$\Delta t = (t_{PKIKP4} - t_{PKIKP2})_{obs} - (t_{PKIKP4} - t_{PKIKP2})_{pred}. \quad (3)$$

We use the bootstrap method (Figure S9) to estimate the uncertainty of the residual
 measurements for the network configuration with 5000 shuffles of the stations in the stacked
 waveforms with replacements, whose mean and standard variation are measured (shown ~~in the~~
 ~~bottom left corners in on top of~~ all panels of Figure 6).

Commented [SP20]: This new figure is supplemented to demonstrate the robustness of bootstraoping method and the advantages of PKIKP multiples in sampling the IC.

Response to Reviewer 3, points #1 and #13.

Commented [SP21]: Correction according to Reviewer 3, point #12.

**Figure 6. Measurements of travel time residuals by cross-correlating stacked waveforms.**

In each panel, solid lines are linearly stacked waveforms of the exotic pairs of PKIKP multiples.

Dashed lines show cross-correlated and shifted original waveforms. Panel titles show the origin

386 times and the measured residuals with bootstrapped uncertainties.

The measured travel time residuals are then converted into relative perturbations from the
 background velocity,

$$\frac{\Delta v}{v} = - \frac{\Delta t}{(\tau_{PKIKP4(n+2)} - \tau_{PKIKP2(n)})_{pred}} \quad (4)$$

In Equation 4, $\tau_{PKIKP4} - \tau_{PKIKP2}$ (or $\tau_{PKIKP3} - \tau_{PKIKP1}$) are the theoretical or observed
 differential travel times $\tau_{PKIKP(n+2)} - \tau_{PKIKP(n)}$ are the theoretically predicted differences of
 propagating times, $\tau_{PKIKP(n)}$ ($1 \leq n \leq 4$), of the IC segments in the PKIKP multiple ray paths,
 and Δt is the measured residual. Thus, the velocity perturbation, $\Delta v/v$, defined in this fashion,
 is compatible with similar quantities inferred for PKP waves traditionally used in other inner
 core studies.

To retain the slight variation in the direction dependence of velocity perturbation, we only
 collect measurements of the PKIKP4-PKIKP2 residuals for the podal configuration, where
 epicentral distances are smaller than 540° , and the PKIKP3-PKIKP1 residuals for the antipodal
 configuration, where epicentral distances are larger than 16055° . These distance criteria help
 reduce the variety of ξ -angles for individual sampling legs of higher multiple arrivals (see
 Figures 1B and 1C). This is somewhat similar to the approach implemented by Costa de Lima
 et al.³², where they used the travel times of the I2* correlation feature, manifested in the
 correlation of much higher-order reverberating PKIKP waves in late earthquake coda, to
 constrain the Earth's inner core anisotropy. Because of these criteria, we retain differential
 residual measurements for 16 events recorded by regional seismic networks in Alaska, the
 mainland United States, and Europe (see Figure 6 for their stacked waveforms and residual
 measurements and Figure S4 for their location maps).

We use a representative angle, ξ' , to represent the sampling direction of an exotic pair to the
 Earth's rotation axis (ERA). For the PKIKP4-PKIKP2 pair, $\xi' = \frac{\xi_1 + \xi_2}{2}$, where ξ_1 and ξ_2 are the

Commented [SP22]: Correction for an error in the original equation. See more details in the response to Reviewer 1, point #7.

angles to the ERA of the forward and backward PKIKP2's legs (see Figure 1B). Similarly, for
 the PKIKP3-PKIKP, the ξ' is the angle relative to the ERA of PKIKP (see Figure 1C). The
 distance criteria mentioned above ensure that the most significant deviation of the
 representative angle ξ' to any of the individual PKIKP legs in the multiples is less than 20° for
 PKIKP or PKIKP2 and less than 10° for PKIKP3 and PKIKP4. Furthermore, because the
 residuals are measured on stacked waveforms over a seismic array, the variation of ξ' for all
 elements must be considered as the uncertainty of the sampling direction. ~~For mathematical~~
 ~~convenience, later used in Equation 1, we calculate the mean and standard deviation of $\cos^2 \xi'$~~
 ~~rather than ξ' .~~ We calculate the mean and standard deviation of $\cos^2 \xi'$ instead of angle ξ' ,
 because Equation 1 is a quadratic function of $\cos^2 \xi'$ and it is mathematically convenient when
 estimating its parameters.

Commented [SP23]: Revisions according to Reviewer 1, point #11.

$$\overline{\cos^2 \xi'} = \frac{1}{N} \sum_{i=1}^N \cos^2 \xi'_i ; \quad (5)$$

$$\Delta \cos^2 \xi' = \sqrt{\frac{1}{N-1} \sum_{i=1}^N (\cos^2 \xi'_i - \overline{\cos^2 \xi'})^2}.$$

The subscript i denotes individual array elements. The measured velocity perturbations ($\Delta v/v$),
 sampling direction ($\cos^2 \xi'$), and their associated uncertainties are plotted as dark squares with
 error bars in Figure 3. ~~Note that the representative uncertainty of the sampling direction, $\Delta \xi'$,~~
 ~~corresponds to the estimated uncertainty, $\Delta \cos^2 \xi'$.~~

Commented [SP24]: We confirm the relation between $\Delta \xi'$ and $\Delta \cos^2 \xi'$ according to Reviewer 1, point #10.

Estimate of anisotropic model

In this study, we use the orthogonal distance regression method⁵⁰ to estimate the anisotropic
 parameters (Equation 1) because this method can account for the measurement uncertainties of
 both ~~dependent and independent parameters~~ explanatory, $\cos^2 \xi'$, and response, $\Delta v/v$, values,
 which are significant in our observed data. The outputs consist of the three anisotropic
 parameters' mean values and their correlated uncertainty in the form of a 3x3 symmetric

covariance matrix. When mantle heterogeneities are corrected for using the DETOX-P3
model⁵³, estimated values for the bulk inner core model parameters are:

$$\begin{array}{l} \varepsilon \quad 1.0 \\ \sigma = -8.3 \\ \gamma \quad 1.7 \end{array} \quad \text{and} \quad C = \begin{array}{ccc} 0.036 & -0.018 & -0.009 \\ -0.018 & 0.576 & -0.118 \\ -0.009 & -0.118 & 0.029 \end{array} \quad (6)$$

Similarly, the estimated values for the 650-km-radius IMIC, given the outer IC is accounted
for using Stephenson et al.'s model³⁴, are:

$$\begin{array}{l} \varepsilon \quad 0.5 \\ \sigma = -15.6 \\ \gamma \quad 3.4 \end{array} \quad \text{and} \quad C = \begin{array}{ccc} 0.036 & -0.018 & -0.009 \\ -0.018 & 0.576 & -0.118 \\ -0.009 & -0.118 & 0.029 \end{array} \quad (7)$$

In Equations 6 and 7, the parameter σ in both cases has large negative values indicating that
the slow directions deviate from the equatorial plane.

Code Availability

The codes used in this study to construct global stacks of the direct wavefield, measuring
residual travel times between pairs of PKIKP multiples and modeling IC cylindrical anisotropy
can be accessed publicly at DOI: 10.5281/zenodo.7317680.

**Data Availability**

We use Obspy and obspyDMT packages^{67,68} to retrieve and process waveform data in this
study. Waveforms data and related metadata were accessed through the following data centers,
IRIS Data Management Center (<http://service.iris.edu>), ETHZ (<http://eida.ethz.ch/fdsnws>),
INGV (<http://webservices.ingv.it>), ORFEUS Data Center (<http://www.orfeus-eu.org>) and
GEOFON Program, GFZ (<http://geofon.gfz-potsdam.de>).

Acknowledgments

The authors are grateful to V. Cormier for helping reproduce Figure S3 from ref.²⁴. This work
was supported by the Australian Research Council Discovery Proposal DP220102815. This
research was undertaken with the assistance of resources and services from the National
Computational Infrastructure (NCI), supported by the Australian Government. We thank three
anonymous reviewers for constructive comments that have greatly improved the final
manuscript.

Commented [SP25]: New numerical experiments were undertaken on the NCI facilities.

[revised manuscript text omitted]

(1996).
- 70. Obayashi, M. *et al.* Finite frequency whole mantle P wave tomography: Improvement of subducted slab
images. *Geophys. Res. Lett.* **40**, 5652–5657 (2013).

Supporting Information for

**Observations of up-to-fivefold reverberating waves through the Earth's center:
distinctly anisotropic innermost inner core**

Thanh-Son Phạm¹ and Hrvoje Tkalčić¹

¹Research School of Earth Sciences, The Australian National University, Canberra ACT, Australia

Contents of this file

Text S1

Figures S1 to S138

Introduction

Text S1 present numerical experiments demonstrating the sensitivities of the long-period signals.

Supplementary figures S1 to S89 support the points from the main text.

Supplementary figures S10 to S13 are mentioned in Text S1.

In addition, we supplement figures featuring 16 sets of differential travel time measurements for the pairs of PKIKP multiples (similar to Figures 5 and S9) in a separate zip file.

All references mentioned in this text are already cited in the main text.

Finite-frequency simulations

We perform full-waveform experiments to (i) benchmark the sensitivity of the exotic reverberating arrivals to the IMIC, (ii) demonstrate the theoretical accuracy, and (iii) identify limitations of the differential travel time measurements between pairs of exotic PKIKP multiples in probing the anisotropic strength of the IMIC.

Experiment setup

We use the spectral element method in 2D (e.g., Komatitsch et al., 2002) to synthesize waveforms in an Earth's cross-section, described by the ak135 model's elastic properties. In the *specfem2d* package, a built-in mesh generator for an Earth's cross-section is provided as an example (global Earth ak135), and we determine mesh sizes to achieve the minimum period of ~7.3 seconds. An isotropic source is located at 200 km depth, and two receiver arrays are equally spaced to span 0–50° and 155–180° epicentral distance ranges. In each range, there are 101 receivers, so the inter-receiver intervals are 0.5° and 0.25°, respectively.

An innermost inner core (IMIC) in the Earth model is introduced by a relative increase of P-wave speed within the centermost 650-km radius from the background model (see a simulation snapshot in Figure S10). We use different isotropic IMICs to simulate the IC probes from different angles with respect to the Earth's rotation axis. Synthesized waveforms, the output of the finite-frequency simulations, are filtered in the same way as the real waveforms, more specifically, bandpass-filtered between 7–13 seconds using a Butterworth filter, two passes, and three corners. In Figures S11 and S12, the waveforms are windowed around the predictions of the PKIKP multiples.

Sensitivity to a 650-km-radius IMIC

Firstly, we examine synthetic waveforms from the original ak135 model (Kennett et al., 1995) to warrant the use of ray-theoretical predictions as a reference for residual travel time measured at 7–13 s (Equation 3 in the main text). Synthetic waveforms of the original ak135 model align well with the ray-theoretical predictions (Figures S11A and S12A). Also, their stacked waveforms of both PKIKP3-PKIKP and PKIKP4-PKIKP2 pairs overlap (Figures S11B and S12B). Thus, we confirm the agreement between finite-frequency and ray-theoretical results.

To prove the sensitivity of the podal and antipodal PKIKP multiples to a 650-km-radius IMIC, we compare the synthetic waveform of the original ak135 model with a model with the IMIC's P-wave speed increased by 3%. In Figures S11C and S12C, there are gradual discontinuities at ~16° and 32° in the plots corresponding to epicentral distances at which PKIKP and PKIKP2 ray paths graze the IMIC boundary. Such discontinuities for PKIKP3 and PKIKP4 are beyond the observed distance range, but due to the faster IMIC, their stacked waveforms arrive early compared to their PKIKP and PKIKP2 counterparts (see Figures S11D and S12D). This experiment benchmarks the exclusive sensitivity of the PKIKP multiples to the IMIC in the 7–13 s period band.

Measurement robustness and limitation

We qualitatively assess the accuracy of the residual measurements by cross-correlating PKIKP multiple waveforms in several epicentral brackets. The IMIC strength varies from 0% (the original model) to 5% in a series of waveform simulations. Because the IMIC radius is around 50% of the IC, the observed P-wave speed increase averaged for the entire IC (Equation 3 in the main text) is ~50% of the input IMIC strength (see the prediction lines in Figures S13A and S13B). Each observed value is measured by the time shift with respect to prediction using synthetic waveforms stacked in corresponding brackets, which is then converted to relative speed perturbation, $\Delta v/v$, using Equation 4.

As can be seen in Figures S13A and S13B, we have a robust recovery of the input IMIC strength using the PKIKP4-PKIKP2 pair in the podal setting (<50°) and the PKIKP3-PKIKP pair in the antipodal setting (>155°) for almost all epicentral-distance brackets. However, the observed $\Delta v/v$ markedly over-estimate the theoretical prediction in the two epicentral distance brackets of 30–40° and 40–50° for PKIKP4-PKIKP2, which is due to large discrepancy of PKIKP2 and PKIKP4 ray paths (see Figure 1B). The worst-case scenarios apply to two analyzed events, 20100812 and 20120930 (see Figure S13C). However, it is noted that when the actual observed value of $\Delta v/v < \sim 1.5\%$ (see Figure 3A), the observation is sufficiently close to the theoretical prediction (see Figure S13A).

Thus, overall, the sensitivity numerical simulations attest to the robustness of the differential travel time measurements presented in this study.

References

- Kennett, B. L. N., & Gudmundsson, O. (1996). Ellipticity corrections for seismic phases. *Geophysical Journal International*, 127(1), 40–48. <https://doi.org/10.1111/j.1365-246X.1996.tb01533.x>
- Kennett, B. L. N., Engdahl, E. R., & Buland, R. (1995). Constraints on seismic velocities in the Earth from traveltimes. *Geophysical Journal International*, 122(1), 108–124. <https://doi.org/10.1111/j.1365-246X.1995.tb03540.x>
- Komatitsch, D., Ritsema, J., & Tromp, J. (2002). The Spectral-Element Method, Beowulf Computing, and Global Seismology. *Science*, 298(5599), 1737–1742. <https://doi.org/10.1126/science.1076024>
- Li, C., van der Hilst, R. D., Engdahl, E. R., & Burdick, S. (2008). A new global model for P wave speed variations in Earth's mantle. *Geochemistry, Geophysics, Geosystems*, 9(5). <https://doi.org/10.1029/2007GC001806>
- Li, X., & Cormier, V. F. (2002). Frequency-dependent seismic attenuation in the inner core, 1. A viscoelastic interpretation. *Journal of Geophysical Research: Solid Earth*, 107(B12), ESE 13-1-ESE 13-20. <https://doi.org/10.1029/2002JB001795>
- Simmons, N. A., Myers, S. C., Johannesson, G., & Matzel, E. (2012). LLNL-G3Dv3: Global P wave tomography model for improved regional and teleseismic travel time prediction. *Journal of Geophysical Research: Solid Earth*, 117(B10), B10302. <https://doi.org/10.1029/2012JB009525>

Supplementary figures

Figure S1. Global stack examples express clear exotic podal and antipodal reverberations. Panel titles note event origin times.

Figure S2. Theoretical reflection coefficients at major Earth's internal interfaces as function of epicentral distances: upper reflection at CMB (PcP), upper reflection at ICB (PKiKP), under-side reflection at ICB (PKIKP), under-side reflection at the CMB (PKIKKIKP), under-side reflection at the Earth's surface (PKIKPPKIKP).

Figure S3. Attenuation operators using the absorption band method (Li & Cormier, 2002) convolved with a given wavelet having 10 s central period. The travel distance is comparable to the diameter of the inner core. Amplitudes are normalized to the incident wavelet. τ_1 represents the lower end of the absorption band, while q_1 is the inverted value of the quality factor Q_1 corresponding to τ_1 . (Readers are referred to the original reference for the complete mathematical representation). Unless the extreme value is used for τ_1 , the long-period signal (10 s) only weakly attenuates while propagating in the inner core.

Figure S4. Location maps for 16 high-events satisfying the epicentral conditionshigh-quality events producing measurements of differential travel times for arrays in the podal and antipodal distance ranges (<540° for PKIKP4-PKIKP2 pairs, and >15560° for PKIKP3-PKIKP4 pairs). Events are denoted by red stars. Seismic stations are shown by blue triangles.

Commented [SP1]: Response to Reviewer 1, point #9 and Reviewer 3, point ...

Figure S5. Observations of PKIKP₂ and PKIKP₃₄ phases in the seismic wavefield from Mw 6.9 Tonga region, 19/06/2019. (A) Seismic records from the Alaskan network are aligned predictions of PKIKP1 arrivals, corrected by the Earth's ellipticity (Kennett & Gudmundsson, 1996). The waveforms are bandpass filtered in the period band of 7–13 seconds. (B) Linear stack of individual waveforms. (E) The spectrograms of stacked waveforms ~~prior to~~ before filtering show the frequency content variation as a function of time. (D, E, F) Similar to (A), (B), (E) but for the PKIKP₃ arrivals.

Commented [SP2]: Fixing label errors of sub-panels.

Figure S6. Cylindrically anisotropic model of Earth's inner core inferred from exotic PKIKP multiples. In the left-hand-side panel, measurements and anisotropic models are plotted as a function of ξ , the angle of the ray path to the rotation axis. The mathematically convenient dependent variable, $\cos^2 \xi$, is used on horizontal axes in the right-hand-side panels. In all panels, differential residual measurements, where mantle heterogeneities are corrected for using the LLNL_G3Dv3 model (Simmons et al., 2012), with associated uncertainties, are plotted by dark squares with error bars. Dark solid lines are the optimal anisotropic models parameterized in Equation 4, and light blue opaque lines represent the uncertainty surrounding the optimal model. Various broken lines show models from previous studies (see the legends). The top

Commented [SP3]: Adding error bars on the sampling directions (left panels).

row (A) compares our inferred bulk IC model, while the bottom row (B) compares our inferred IMIC model with its predecessors.

Figure S7. Similar to Figure S6, mantle heterogeneities are corrected for using the MIT-P08 model (C. Li et al., 2008).

Figure S8. Similar to Figure S6, but mantle heterogeneities are not considered in this case.

Figure S9. Measuring residual travel times of PKIKP4-PKIKP2 observation for the Mw 7.1 Anchorage earthquake 30/11/2018 by stacked waveform cross-correlation. A) The histogram shows the bootstrapping result comprising of residual travel times for 5000 resampled station sets with replacement. The estimated mean and standard deviation are written on the top left. B) The Earth cross-section compares how the IC is sampled by the exotic PKIKP multiples with 'traditional' PKP waves along the SSI-Alaska path. C) The stacked PKIKP2 and PKIKP4 waveforms are plotted using their actual times with respect to the theoretical predictions. D) Similar to C) but the stacked PKIKP4 waveform is aligned to PKIKP2 by the estimated mean residual.

Figure S10. A snapshot at 650 seconds of the `specfem2d` (e.g., Komatitsch et al., 2002) simulation in an Earth's cross-section. A 200-km-deep seismic source is marked by an orange cross; podal and antipodal seismic arrays are denoted by green triangles. Negative and positive z-velocity components are plotted in blue and red colors. A 650-km-radius IMIC is characterized by an increase from the ak135 background model.

Figure S11. *A)* Individual synthetic waveforms of PKIKP and PKIKP3 (bandpass-filtered in the band 7–13 s) are aligned with respect to their ray-theoretical predictions. The main arrivals are located near 0 s, while the depth reflections are seen ~43 s corresponding to a 200-km-deep source. We use the ak135 model (Kennett et al., 1995) for both the computational domain and travel time prediction. *B)* Black (PKIKP) and gray (PKIKP3) waveforms are stacked from the individual waveforms in *A)*. Stacked PKIKP3 waveform (gray) is plotted in both left and right panel. *C)* and *D)* are similar to *A)* and *B)* but for the model with an IMIC P-wave speed increased by 3%.

The stacked PKIKP3 waveform (gray dashed line) in the left panel of D) is identical to that in E) but shifted to align with the stacked PKIKP1 waveform (black line), because PKIKP3 waves arrive earlier than predicted due to the faster IMIC in this model.

Figure S12. Similar to Figure S11, but for the PKIKP2 and PKIKP4 phases.

Figure S13. Method sensitivity test to recover the increased P-wave speed confined in a 650-km-radius IMIC using PKIKP4-PKIKP2 pair (panel A) and PKIKP3-PKIKP1 pair (panel B). The horizontal axes represent a synthetic input increase in the IMIC's P-wave speed, $\Delta v/v$. Vertical axes represent the IC-average P-wave speed increases obtained by cross-correlating synthetic waveforms in five distance brackets (see the legends) in the same way the observations are made. The black solid lines show the predicted (theoretical) values corresponding to the inputs. C) Sub-panels show the distribution of epical distances corresponding to 16 residual measurements (see

Figure S4. The histograms are colored by distance brackets in the same convention as the above panels.

REVIEWERS' COMMENTS

Reviewer #1 (Remarks to the Author):

I thank the authors for addressing all my concerns carefully. Appropriate details and corrections to text as well as figures have been added, which helps strengthen the robustness of scientific approach in this study. Authors make a noteworthy observation of an exotic IC phase, while adding to the current understanding on inner most IC. Harnessing as much information from indiscernible phases are valuable in expanding the sampling methods of deep Earth. Sets good precedent for future IC probing. I recommend the article for publication.

Wish you the best!

Reviewer #2 (Remarks to the Author):

The authors have conducted nice forward modeling (compute synthetic seismograms) to benchmark their study. I am quite convinced about their results after revision, and I am happy that all my questions have been well addressed.

I would like to recommend the paper to be accepted by NC.